# RobustLight++: A Meta-Diffusion Framework for Robust Traffic Signal Control

## Abstract

Despite remarkable progress in Reinforcement Learning (RL) for Traffic Signal Control (TSC), existing methods largely lack the ability to generalize across cities, limiting their applicability in real deployments. The recent SoTA method RobustLight improves robustness but still exhibits weak transfer performance, high inference latency, and limited resistance to sensor failures. In this paper, we present RobustLight++, a meta-diffusion-based framework designed to explicitly learn transferable representations among heterogeneous urban environments. By theoretically linking DDIM with Reptile meta-learning, RobustLight++ enables a diffusion policy that supports both zero-shot deployment and few-shot adaptation in unseen cities, significantly reducing the cost of retraining and data collection in new domains. Comprehensive experiments on large-scale real-world benchmarks demonstrate superior cross-city transfer capability, with performance gains ranging from 7.41% to 52.13% under diverse noise conditions, and consistent improvements over all competing baselines in unseen environments. In addition, RobustLight++ achieves up to 91.9% reduction in inference latency, ensuring real-time applicability. The proposed framework delivers a practical solution toward scalable, transferable, and robust urban traffic control systems. Our code is available at `https://anonymous.4open.science/r/RobustLightPlus-E14F`.

## 1 Introduction

Traffic Signal Control (TSC) is a fundamental component of urban traffic management, aiming to alleviate congestion and improve mobility (Liang et al., 2018). While recent Reinforcement Learning (RL) based approaches have demonstrated promise in adaptive signal control (Wei et al., 2019c), their practical deployment is often hindered by critical challenges in generalization, efficiency, and robustness, particularly when faced with imperfect real-world sensor data. The recent state-of-the-art (SoTA), RobustLight (Li et al., 2025b;a), marks a significant step forward. However, several critical limitations still impede its large-scale, real-world application: 1) its per-city training paradigm, which requires a separate model for each environment, fundamentally restricts its ability to generalize and transfer knowledge across diverse urban settings; 2) its inference and adaptation processes are computationally intensive, resulting in high latency that is prohibitive for real-time control applications; and 3) although effective in controlled research settings, it falls short of meeting industrial-grade standards under the complex, dynamic, and noisy conditions of real-world traffic. These limitations underscore the urgent need for a more generalizable, efficient, and robust framework capable of adapting to new environments with minimal overhead.

To address the challenge of generalization, meta-learning frameworks have been proposed (Huang et al., 2021; Zang et al., 2020) to extract transferable knowledge across different cities. Concurrently, denoising diffusion models, such as Denoising Diffusion Implicit Model (DDIM) (Song et al., 2020a), have emerged as exceptionally powerful generative tools. However, their potential for robust decision-making and meta-generalization in RL remains largely underexplored.

In this work, we bridge this gap by introducing **RobustLight++**, a novel meta-diffusion framework. We establish a key theoretical connection between the iterative sampling process of DDIM and the meta-update rule of Reptile (Nichol et al., 2018). Our core insight is that the DDIM sampling process naturally mirrors a gradient-based meta-optimization, where the learned noise predictor implicitly

guides the policy toward a robust solution space. Based on this connection, we propose a unified framework that recasts the diffusion model as a powerful meta-learner, enabling rapid and robust policy adaptation across cities with only a few fine-tuning steps.

This work significantly narrows the gap between academic research and the practical demands of industrial-scale traffic optimization. Our primary contributions are:

- We establish a novel theoretical connection between the DDIM sampling process and the Reptile meta-learning, framing DDIM as a learnable and robust meta-optimizer for policy adaptation.

- We propose **RobustLight++**, a unified meta-diffusion framework that leverages this connection for cross-city TSC, enabling superior generalization and rapid adaptation with a single shared model.

- We validate the effectiveness of our approach on multiple real-world benchmarks, where it achieves SoTA performance, reduces inference latency by over 91.9%, and demonstrates industrial-grade availability.

## 2 RELATED WORKS

TSC has progressed from rule-based methods (Webster, 1958) to adaptive systems like SCOOT (Hunt et al., 1981), SCATS (Lowrie, 1990), and RHODES (Mirchandani & Head, 2001), which reduce delays dynamically. Control-theoretic methods such as Max Pressure (Varaiya, 2013) and Efficient Pressure (Wu et al., 2021) offer robustness but rely on simplified assumptions. Reinforcement learning has become the dominant paradigm (Wei et al., 2021), with approaches like IntelliLight (Wei et al., 2018), PressLight (Wei et al., 2019a), CoLight (Wei et al., 2019b), and MetaLight (Zang et al., 2020) achieving strong results. Recent works explore multi-agent coordination (Song et al., 2024) and multi-modal or hierarchical representations (Yu et al., 2023; Wang et al., 2024; Ruan et al., 2024; Duan et al., 2025), yet most still rely on fixed, hand-crafted state features. Viewing traffic as a complex system (Mitchell, 2009; Strogatz, 2001), hierarchical RL provides scalable solutions (Salehkaleybar et al., 2019; Shen et al., 2020).

Meta-learning has advanced through diverse strategies, including latent embedding optimization (Rusu et al., 2018), differentiable convex solvers (Lee et al., 2019), implicit gradients (Rajeswaran et al., 2019; Zhang et al., 2023), and sparsity-aware adaptation (Von Oswald et al., 2021). Studies have also explored the trade-off between rapid learning and feature reuse (Raghu et al., 2019). Recently, MetaDiff (Zhang et al., 2024) introduced a task-conditional diffusion-based framework that generalizes gradient descent with learnable momentum and uncertainty modeling. Meta-learning has recently emerged as a promising direction in TSC for improving generalization and adaptability across varying traffic scenarios. Early works like MetaLight (Zang et al., 2020) and CrossLight (Sun et al., 2024) applied meta-learning for quick adaptation and cross-scenario generalization, but overlooked safety concerns in TSC.

Diffusion models have achieved remarkable success in generative modeling (Ho et al., 2020a; Song et al., 2020a; Nichol & Dhariwal, 2021b), with improvements in training stability and sample quality through advanced beta schedules (Xiao et al., 2021; Nichol & Dhariwal, 2021a). Classifier-free guidance further enhanced controllability (Ho & Salimans, 2022). Recently, diffusion has been extended to RL for robust decision-making under uncertainty, such as DiffLight for missing data in TSC (Chen et al., 2024), RobustLight for policy robustness (Li et al., 2025b), and DMBP for offline RL with noisy states (Yang & Xu, 2023). However, these algorithms suffer from slow inference speed, making them impractical for real-world deployment, and their reconstruction performance remains suboptimal.

## 3 PRELIMINARY

### 3.1 CROSS-CITY FEW-SHOT POLICY TRANSFER

We consider a meta-RL setting for TSC, where $K$ source cities $\mathcal{T}_1^{\text{src}}, \ldots, \mathcal{T}_K^{\text{src}}$ provide offline trajectories, and a target city $\mathcal{T}_{\text{tgt}}$ lacks prior data but allows limited online interactions. The objective is

to transfer a policy pre-trained on $\mathcal{T}_k^{\text{src}}, k = 1, .., K$ and adapt it efficiently to $\mathcal{T}_{\text{tgt}}$ using only a few online rollouts, thereby maximizing travel efficiency with minimal deployment cost.

## 3.2 TRAFFIC SIGNAL CONTROL

We use a four-way intersection, as depicted in Figure 1, to introduce key concepts and definitions for TSC. A road network comprises multiple intersections, each with $N$ road segments, denoted as $\{Inter_1, \ldots, Inter_N\}$. Each intersection is equipped with four directional sensors (e.g., cameras, radars) monitoring three lanes per direction. Sensor states are color-coded as green for normal, orange for noise attacks, and red for sensor damage, as shown in Figure 1(b). A vehicle's path through an intersection, from an entry lane ($lane_{in}$) to an exit lane ($lane_{out}$), is defined as $TM = (lane_{in}, lane_{out})$, as illustrated in Figure 1(c). A traffic signal phase consists of two distinct movements, $TM_i$ and $TM_j$ ($i \neq j$), denoted as $p_w = (TM_i, TM_j)$, as shown in Figure 1(d).

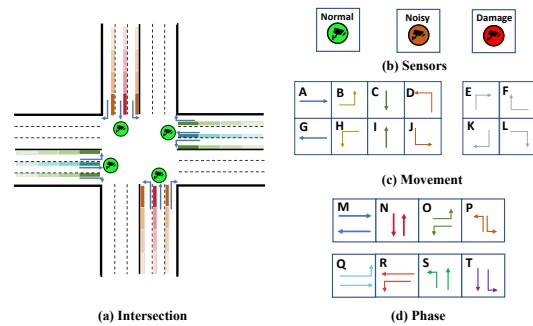

Figure 1: Definition of the TSC. Sensors are used to acquire vehicle information, which is fed into the TSC algorithm to output the appropriate phase, as shown in Figure 1(d).

## 3.3 ADVERSARIAL ATTACKS AND SENSOR DAMAGE

We define four adversarial attacks and physical sensor damage affecting TSC. The **Gaussian Noise Attack** adds Gaussian noise $\mathcal{N}(\mu, \sigma^2)$ scaled by intensity $k$ to the state $s$, yielding $\tilde{s}_t = s_t + k \cdot \mathcal{N}$. **The U-rand Attack** introduces uniform random noise $\mathcal{U}$ within intensity $k$, expressed as $\tilde{s}_t = s_t + k \cdot \mathcal{U}(I, I)$, where $I$ is the identity matrix. **The MAD Attack** selects noise within an $\ell_\infty$ ball $\mathbf{B}_d(s, k)$ to maximize policy divergence: $\tilde{s}_t = s_t + \arg\max_{\tilde{s} \in \mathbf{B}_d(s,k)} D(\pi_\phi(\cdot|s) \parallel \pi_\phi(\cdot|\tilde{s}))$. **The MinQ Attack** chooses noise within $\mathbf{B}_d(s, k)$ to minimize the $Q$-value: $\tilde{s}_t = s_t + \arg\min_{\tilde{s} \in \mathbf{B}_d(s,k)} Q(\tilde{s}_t, \pi_\phi(\cdot|\tilde{s}))$. Finally, **physical sensor damage** (e.g., due to weather or human factors) leads to unobserved state dimensions, which we model as $\tilde{s}_t = Mask \cdot s_t$.

## 3.4 DENOISING DIFFUSION IMPLICIT MODELS

Diffusion models generate data by reversing a Markovian noise process over $T$ steps. To accelerate inference, DDIM (Song et al., 2020a) introduces a strided schedule $\tau_1, \ldots, \tau_S$ with $S \ll T$, skipping redundant steps. The transition distribution is reformulated as:

$$\mathbf{x}_{t-1} = \sqrt{\bar{\alpha}_{t-1}} \left( \frac{\mathbf{x}_t - \sqrt{1 - \bar{\alpha}_t} \boldsymbol{\epsilon}_\theta^{(t)}(\mathbf{x}_t)}{\sqrt{\bar{\alpha}_t}} \right) + \sqrt{1 - \bar{\alpha}_{t-1} - \sigma_t^2} \boldsymbol{\epsilon}_\theta^{(t)}(x_t) + \sigma_t \boldsymbol{\epsilon},$$

where $\boldsymbol{\epsilon}_\theta^{(t)}$ predicts the noise at step $t$, and $\sigma_t^2 = \delta \cdot \tilde{\beta}_t$ modulates stochasticity via a tunable hyperparameter $\delta > 0$, $\tilde{\beta}_t = \sigma_t^2 = \frac{1 - \bar{\alpha}_{t-1}}{1 - \bar{\alpha}_t} \cdot \beta_t$, $\alpha_t = 1 - \beta_t$ and $\bar{\alpha}_t = \prod_{i=1}^t \alpha_i$ and $\sigma \sim \mathcal{N}(0, I)$. Setting $\delta = 0$ yields a deterministic generation process. DDIM preserves the marginal distribution of DDPM (Ho et al., 2020b) but allows for significantly faster sampling. To capture the underlying data geometry, Song et al. (2020b) approximate the true score $\nabla \log p_t(\mathbf{x})$ using a parameterized estimator $\mathbf{s}_\theta(\mathbf{x}, t)$. The model parameters $\theta$ by minimizing the weighted Fisher divergence, which takes the form of a weighted Mean Squared Error (MSE):

$$J(\theta; \lambda) := \frac{1}{2} \int_0^T \lambda(t) \mathbb{E}_{p_t} \left[ \| \nabla \log p_t(\mathbf{x}) - \mathbf{s}_\theta(\mathbf{x}, t) \|^2 \right] dt. \tag{1}$$

Here, $\lambda(t) > 0$ is a time-dependent weighting function that balances the learning signal across different noise scales. Kwon et al. (2022) estimate the Wasserstein distance $W_2(p_0, q_0)$ between a given data distribution $p_0$ and the marginal of the generated samples $q_0$ obtained by the score-based model $\mathbf{s}_\theta(\mathbf{x}, t)$.

## 3.5 REPTILE META LEARNING

Reptile is a first-order gradient-based meta-learning algorithm, similar in spirit to Model-Agnostic Meta-Learning (MAML) (Finn et al., 2017) but computationally more efficient. It iteratively samples tasks, performs $k$ steps of task-specific Stochastic Gradient Descent (SGD) from an initialization $\theta$ to obtain $\phi = \nabla \mathcal{L}_{\mathcal{T}}(\theta)$, and updates $\theta$ toward $\phi$:

$$\theta \leftarrow \theta - \mu(\theta - \phi), \tag{2}$$

where $\mu$ is the meta step size. This process encourages $\theta$ to lie close to the optimal manifold $\mathcal{M}_i$ of each task $\mathcal{T}_i$. Assuming each task $\mathcal{T}_i$ has an optimal parameter manifold $\mathcal{M}_i(\phi)$, the learning objective becomes minimizing the aggregated distance $|\theta - \mathcal{M}_i(\phi)|_2^2$ across tasks. The Reptile update approximates the gradient of this objective by treating $\phi$ as a proxy for the projection of $\theta$ onto $\mathcal{M}_i(\phi)$:

$$\nabla_\theta \big[ \frac{1}{2} |\theta - \mathcal{M}_i(\phi)|_2^2 \big] \approx (\theta - \phi). \tag{3}$$

This enables generalization by locating $\theta$ near the intersection of task-specific optima, facilitating fast adaptation in new tasks.

# 4 METHODS

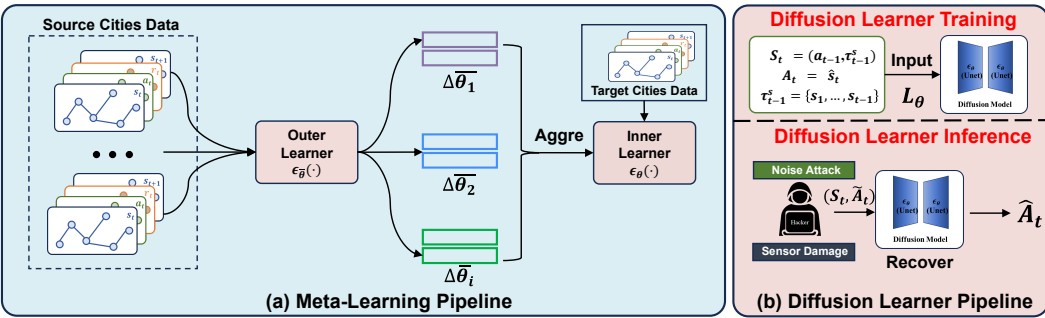

(a) Meta-Learning Pipeline

(b) Diffusion Learner Pipeline

Figure 2: Overall framework of RobustLight++. In meta-learning pipeline, it employs Reptile meta-learning with a two-level architecture. The outer diffusion learner acquires meta-parameters from source cities data, while the inner diffusion learner performs final parameter adaptation on target cities data. In the diffusion pipeline, it uses a trained model to recover the state.

## 4.1 DDIM WITH REPTILE META LEARNING THEORY

We present a unified view that connects the DDIM with gradient-based meta-learning, particularly the Reptile meta learning. By reformulating the DDIM sampling process, we reveal its structural equivalence to a generalized meta-update rule with momentum and uncertainty modeling. The DDIM sampling step from time $t$ to $t-1$ can be written as:

$$\mathbf{x}_{t-1} = \frac{\sqrt{\bar{\alpha}_{t-1}}}{\sqrt{\bar{\alpha}_t}} \mathbf{x}_t - \left( \frac{\sqrt{\bar{\alpha}_{t-1}}\sqrt{1-\bar{\alpha}_t}}{\sqrt{\bar{\alpha}_t}} - \sqrt{1-\bar{\alpha}_{t-1}-\sigma_t^2} \right) \boldsymbol{\epsilon}_\theta^{(t)}(\mathbf{x}_t) + \sigma_t \boldsymbol{\epsilon}. \tag{4}$$

By defining the following time-dependent parameters:

$$\gamma = \frac{\sqrt{\bar{\alpha}_{t-1}}}{\sqrt{\bar{\alpha}_t}}, \quad \xi = \sigma_t, \quad \eta = \frac{\sqrt{\bar{\alpha}_{t-1}}\sqrt{1-\bar{\alpha}_t}}{\sqrt{\bar{\alpha}_t}} - \sqrt{1-\bar{\alpha}_{t-1}-\sigma_t^2}.$$

### 4.1.1 LINKING TO REPTILE META-LEARNING.

We begin by observing the structural similarity between the update rule in diffusion-based models and classical gradient-based optimization (Zhang et al., 2024). Consider the standard gradient descent formulation:

$$\theta \leftarrow \theta - \eta \nabla \mathcal{L}(\theta), \tag{5}$$

where $\theta$ denotes the model parameters, $\nabla\mathcal{L}(\theta)$ is the loss function, and $\eta$ is the learning rate. This iterative update aims to minimize the loss by moving in the direction of the negative gradient. In comparison, the deterministic update rule in DDIM can be reformulated as:

$$\mathbf{x}_{t-1} = \mathbf{x}_t - \eta\boldsymbol{\epsilon}_\theta^{(t)}(\mathbf{x}_t) + (\gamma - 1)\mathbf{x}_t + \xi\boldsymbol{\epsilon}, \tag{6}$$

where $x_t$ denotes the generated sample at time step $t$, $\boldsymbol{\epsilon}_\theta^{(t)}$ is the learned noise predictor, and the additional terms $(\gamma - 1)\mathbf{x}_t$ and $\xi\boldsymbol{\epsilon}$ respectively introduce momentum and stochasticity into the update dynamics. By comparing Eq. (5) with Eq. (6), it becomes evident that DDIM can be interpreted as performing a noise-conditioned, time-varying descent in the data space. The term $\boldsymbol{\epsilon}_\theta^{(t)}$ serves as a surrogate gradient, while $\eta$ remains a scaling factor modulating the update magnitude. This interpretation naturally bridges the diffusion process with meta-learning. In particular, we draw parallels to the Reptile algorithm, a first-order meta-learning method that updates the meta-parameters $\theta$ by moving them toward task-specific adapted weights $\phi$, which are obtained by applying several gradient steps on a sampled task. Formally, Reptile performs:

$$\theta \leftarrow \theta + \mu(\phi - \theta), \tag{7}$$

where $\phi$ approximates $\theta - \eta\nabla\mathcal{L}_\mathcal{T}(\theta)$ ($\mathcal{T}$ represents the specific task dataset) after a few steps gradient updates. In this light, the DDIM update can be viewed as an implicit meta-update, where $x_t$ plays the role of the initialization $\theta$, and $\boldsymbol{\epsilon}_\theta^{(t)}$ mimics a task-specific gradient.

Beyond this structural resemblance, DDIM incorporates additional capabilities absent in vanilla Reptile. The momentum-like term $(\gamma - 1)x_t$ introduces time-dependent inertia that enhances stability and adaptivity during sampling, akin to a learnable momentum coefficient. Meanwhile, the injected Gaussian noise $\xi\boldsymbol{\epsilon}$ serves as a form of stochastic regularization, promoting robustness and avoiding overfitting to any single generative trajectory.

DDIM extends beyond standard gradient descent by naturally incorporating ideas from meta-optimization frameworks such as Reptile. A key advantage of DDIM is its ability to derive important hyperparameters like $\gamma$ and $\xi$ analytically from the diffusion schedule, which removes the need for manual tuning. This positions DDIM as a principled and efficient meta-optimization method that supports few-shot learning in both generative and discriminative tasks.

### 4.2 FRAMEWORK OVERVIEW

We propose a meta-learning framework that unifies DDIM with Reptile updates for cross-city TSC. In the meta learning pipeline, each city is treated as a task in a multi-task paradigm: an *outer loop* aggregates information across source cities and updates a shared initialization, while an *inner loop* fine-tunes that initialization on a target city. In the diffusion pipeline, we use the trained model of meta learning to recover the state. Crucially, both loops optimize the same diffusion-based loss, ensuring consistency between generalization and specialization phases.

#### 4.2.1 META LEARNING PIPELINE.

Our diffusion learner is trained via offline meta-learning over a collection of multi-city datasets. Specifically, we first collect logged trajectories $\mathcal{D}_\tau = \{(s_t, a_t, s_{t+1})\}$ from each training city $\tau$, and aggregate them into a global offline corpus $\mathcal{D}_{\text{meta}}$. During meta-training, the diffusion model is optimized to learn a cross-domain state distribution by minimizing the diffusion loss (Yang & Xu, 2023):

$$L_{\text{diff}}(\theta; \mathcal{T}_i) = \mathbb{E}_{i\sim\mathcal{U}_K, \epsilon_t\sim\mathcal{N}(0,I), (s_{t-N},...,s_{t+M-1})\in D_\nu}$$

$$\left\|\epsilon_\theta(\tilde{s}_t^i, c_{t-1}, i) - \epsilon_t^i\right\|_2 + \sum_{m=t+1}^{t+M-1}\left\|\epsilon_\theta(\tilde{s}_m^i, \hat{c}_{m-1}, i) - \epsilon_m^i\right\|_2, \tag{8}$$

Where the condition represents as $c_t = (a_{t-1}, \tau_{t-1}^s)$, $a_{t-1}$ is the previous TSC action, $\tau_{t-1}^s = \{s_1, ..., s_{t-1}\}$ is the TSC state trajectory, $\hat{c}_{m-1} = (a_{m-1}, \tau_{m-1}^{\hat{s}})$, and $\tau_{m-1}^{\hat{s}}$ is the predicted state trajectory. This loss balances immediate and future timesteps by penalizing the mismatch between predicted and true noise across a window of length $N + M$.

We use Reptile to meta-learn an initialization $\theta$ that performs well on any city after a few updates. For each outer iteration with $N$ source tasks, we compute:

$$\phi = \theta - \eta \nabla_\theta L_{\text{diff}}(\theta; \mathcal{T}_i), \quad \theta \leftarrow \theta + \frac{\mu}{N} \sum_{i=1}^{N} (\phi - \theta), \tag{9}$$

where $\mu$ is the outer-loop step size. This first-order update aligns the global parameters toward the average task-adapted parameters. During meta-training, we sample source tasks (e.g., $Hangzhou$) and run $k$ diffusion-loss inner updates to compute each $\phi$. The aggregated update $\theta$ refines the shared initialization. During meta-training, the denoising network is optimized on offline data from multiple source cities without environment interaction. After convergence, the diffusion model is frozen and deployed for state recovery in downstream control. At test time, the trained model supports zero-shot state reconstruction from noisy inputs. When transferring to a target city (e.g., $JiNan$), we further perform few-shot adaptation by fine-tuning $\theta$ with the diffusion loss (Eq. 8) to improve reconstruction fidelity and generalization. The algorithm is shown in Algorithm 1. Our algorithm convergence demonstration is in theorem 1.

**Theorem 1.** *Let $L_{\text{diff}}$ be the score matching loss. The Wasserstein-2 distance between the target data distribution $p_0$ and the model distribution $q_0$ is bounded by:*

$$W_2(p_0, q_0) \leq \underbrace{\mathcal{C}_{score}\sqrt{L_{\text{diff}}(\theta)}}_{\text{Score Error}} + \underbrace{\mathcal{C}_{disc} \cdot \mu}_{\text{Reptile Error}} + \underbrace{\mathcal{C}_{init}W_2(p_T, q_T)}_{\text{Initialization}}, \tag{10}$$

*where $\mathcal{C}_{score}$ and $\mathcal{C}_{disc}$ are constants determined by the integrated one-sided Lipschitz coefficients of the vector field.*

**Remark.** Theorem 1 indicates that the generation quality is governed by two key factors: 1) the accuracy of the surrogate gradient (score matching loss), and 2) the step size $\mu$ of the Reptile meta-update. The one-sided Lipschitz (Kwon et al. (2022)) ensures that the reptile discretization error accumulates linearly ($\mathcal{O}(\mu)$) rather than exponentially, guaranteeing the stability of the meta-trajectory. Detailed proof is in Appendix B.

### 4.2.2 DIFFUSION PIPELINE.

The pipeline starts from *Denoising via DDIM*. During evaluation, TSC sensors may be corrupted by Gaussian noise, MAD, U-rand, or Min-Q perturbations. To restore the true state, we employ the diffusion model meta-trained by Reptile. At each diffusion timestep $j$, the network receives the current noisy observation $\tilde{s}_t^j$, the last estimated state $c_{t-1}$, and the timestep index $j$. The denoising update follows the DDIM rule:

$$\tilde{s}_t^{j-1} = \sqrt{\bar{\alpha}_{j-1}} \frac{\tilde{s}_t^j - \sqrt{1 - \bar{\alpha}_j}\,\epsilon_\theta(\tilde{s}_t^j, c_{t-1}, j)}{\sqrt{\bar{\alpha}_j}} + \sqrt{1 - \bar{\alpha}_{j-1} - \sigma_j^2}\,\epsilon_\theta(\tilde{s}_t^j, c_{t-1}, j) + \sigma_j z, \tag{11}$$

where $z \sim \mathcal{N}(0, I)$. Iterating this process for $j = T, ..., 1$ yields the reconstructed state $\hat{s}_t$, effectively removing adversarial or stochastic corruptions.

Next, *Repainting* is conducted to compensate the missing parts. We adapt conditional diffusion to infer and restore missing or damaged sensor readings. Let $m$ be the binary mask indicating known ($m = 1$) and unknown ($m = 0$) entries in $\tilde{A}_t^j$. At each reverse step, we sample the known portion by forward diffusion:

$$\tilde{s}_{t,\text{known}}^{j-1} = \sqrt{\bar{\alpha}_j}\,\tilde{s}_{t,\text{known}} + \sqrt{1 - \bar{\alpha}_j}\,\epsilon, \tag{12}$$

and recover the unknown portion via the conditional reverse update:

$$\tilde{s}_{t,\text{unknown}}^{j-1} = \sqrt{\bar{\alpha}_{j-1}} \frac{\tilde{s}_{t,\text{unknown}}^j - \sqrt{1 - \bar{\alpha}_j}\,\epsilon_\theta(\tilde{s}_t^j, c_{t-1}, j)}{\sqrt{\bar{\alpha}_j}} + \sqrt{1 - \bar{\alpha}_{j-1} - \sigma_j^2}\,\epsilon_\theta(\tilde{s}_t^j, c_{t-1}, j) + \sigma_j z, \tag{13}$$

where $\sigma$ and $z$ are independent Gaussian noises. The combined sample for the next iteration is

$$\tilde{s}_t^{j-1} = m \odot \tilde{s}_{t,\text{known}}^{j-1} + (1 - m) \odot \tilde{s}_{t,\text{unknown}}^{j-1}. \tag{14}$$

By iterating from $j = T$ down to $j = 1$, the repaint algorithm reconstructs the full state $\hat{s}_t$. The complete procedure is detailed in Algorithm 2.

## 5 EXPERIMENTS

All experiments were conducted on an Ubuntu 22.04 server equiped with 8 × NVIDIA GeForce RTX 4090 GPUs and 512GB DDR5 RAM. However, both the training and inference of our model only require a single RTX 4090 GPU.

### 5.1 DATASETS AND COMPARED METHODS

We evaluate our approach using real-world traffic datasets simulated in Cityflow (Zhang et al., 2019), measuring the Average Travel Time (ATT) over a 60-minute period. The datasets include start/end vehicle points with a fixed motion model, covering 7 traffic datasets from JiNan and HangZhou (China) and New York (Zheng et al., 2019) (USA). JiNan includes 12 intersections ($3 \times 4$ grid) with three datasets: $JiNan_1$, $JiNan_2$, and $JiNan_3$. HangZhou has 16 intersections ($4 \times 4$ grid) and two datasets: $HangZhou_1$ and $HangZhou_2$. New York features a large-scale network with 192 intersections ($28 \times 7$ grid) and two datasets: $Newyork_1$ and $Newyork_2$.

We compare our method against traditional and RL-based TSC approaches. Traditional methods include FixedTime (Webster, 1958) and Advanced-Maxpressure (Zhang et al., 2022). RL-based methods include Advanced-CoLight (Zhang et al., 2022) and Advanced-Mplight (Zhang et al., 2022), which is based on FRAP (Zheng et al., 2019), RobustLight (Li et al., 2025b). Our proposed RobustLight++ integrates both traditional and RL-based methods, enabling real-time data recovery and robust ATT evaluation under various sensor noise and damage scenarios. All results are averaged over ten independent runs. More datasets and compared methods are shown in the **Appendix A**.

### 5.2 RESULTS

This subsection presents the results of our experiments, evaluating RobustLight's performance under various conditions, including resilience to noise attacks and sensor damage, using ATT on real-world traffic datasets.

#### 5.2.1 NOISE ATTACK ON STATE RESULTS

Table 1 presents the ATT performance of different TSC algorithms under multiple noise attack settings on the $JiNan$ and $HangZhou$ datasets. We consider four types of perturbations: Gaussian, U-rand, MAD, and MinQ, with two levels of noise intensity. RobustLight++ consistently improves performance across most of the baselines and noise types, indicating its strong robustness under both stochastic and adversarial attacks. In particular, compared to both traditional and RL-based controllers, RobustLight++ yields substantial gains in high-noise scenarios. These results validate that RobustLight++ not only recovers degraded policies but also establishes a more stable decision process against various noise patterns, highlighting its potential for real-world deployment under imperfect sensing conditions. On average, RobustLight++ achieves 6.77% lower ATT with the most significant improvement reaching up to 26.18% under the MinQ attack in $JiNan_1$ with Advanced-CoLight.

#### 5.2.2 SENSOR DAMAGE ON STATE RESULTS

We further investigate the performance of traditional and RL-based TSC algorithms under deliberate sensor failures, simulating the loss of information from $sensor_W$ and $sensor_E$ by masking the input data. Table 2 summarizes the ATT values across five datasets, where 25% masking refers to damage in $sensor_W$ and 50% masking simulates failures in both $sensor_W$ and $sensor_E$. To mitigate the resulting observation loss, we apply the Repaint algorithm within RobustLight++ to recover the missing state information. Across all datasets and methods, RobustLight++ achieves an average improvement of 12.75% over RobustLight, Particularly enhancing performance in heavily degraded environments where traditional models collapse, achieving a maximum improvement of 52.13% in $HangZhou_1$ under 50% sensor damage. Notably, even under severe dual-sensor damage (50% mask), RobustLight++ enables most of methods to maintain performance comparable to or better than FixedTime, demonstrating its potential for robust real-world deployment under partial observability. The results show that RobustLight++ consistently improves performance over RobustLight across most of methods and scenarios, demonstrating its strong resilience to sensor failures.

Table 1: **Performance of ATT in $JiNan$, $HangZhou$. Our RobustLight++ recovers the state of traditional and RL-based TSC algorithms.**

| Dataset | Noise Type | Noise Scale | FixedTime base | Advanced-CoLight base | RobustLight | RobustLight++ | Advanced-MpLight base | RobustLight | RobustLight++ |
|---|---|---|---|---|---|---|---|---|---|
| $JiNan_1$ | Gaussian | 3.5 | | 316.96±4.63 | 328.38±3.90 | **282.26±1.35** | 327.93±8.76 | 303.83±5.98 | **289.85±3.76** |
| | | 4.0 | | 329.24±3.73 | 346.47±3.85 | **281.73±1.77** | 338.68±7.41 | 307.19±2.21 | **289.25±1.82** |
| | U-rand | 3.5 | 428.11 | 483.26±74.90 | 365.55±11.95 | **327.33±3.81** | 417.81±12.34 | 362.61±9.16 | 399.88±19.17 |
| | | 4.0 | | 497.32±64.04 | 388.69±25.74 | **329.89±8.66** | 434.84±11.18 | 365.96±7.52 | 377.99±16.26 |
| | MAD | 3.5 | | 454.77±9.38 | 301.29±3.43 | **282.45±2.03** | 339.41±37.82 | 305.16±19.99 | **279.95±1.69** |
| | | 4.0 | | 495.47±15.55 | 305.32±4.13 | **286.63±2.31** | 355.38±36.38 | 313.24±30.45 | **290.00±1.61** |
| | MinQ | 3.5 | | 493.10±96.80 | 388.41±85.49 | **299.67±8.34** | 347.62±28.10 | 297.97±14.57 | **284.94±7.45** |
| | | 4.0 | | 520.99±115.29 | 387.87±62.70 | **286.29±2.57** | 356.30±30.91 | 307.07±21.89 | **288.30±4.54** |
| $JiNan_3$ | Gaussian | 3.5 | | 320.04±11.40 | 277.17±2.40 | **263.89±1.92** | 378.13±32.17 | 279.79±1.79 | **261.91±1.17** |
| | | 4.0 | | 331.64±14.13 | 283.54±4.32 | **259.75±2.31** | 390.93±25.46 | 290.37±2.56 | **270.69±1.45** |
| | U-rand | 3.5 | 383.01 | 481.55±43.52 | 333.46±19.05 | **290.39±1.81** | 496.63±44.43 | 354.45±11.06 | **314.14±5.68** |
| | | 4.0 | | 490.09±44.06 | 344.34±14.80 | **288.89±3.16** | 505.93±41.98 | 363.10±18.75 | **312.17±3.62** |
| | MAD | 3.5 | | 446.89±15.76 | **273.78±4.00** | 281.70±3.29 | 442.19±81.19 | 268.96±11.41 | **259.25±2.87** |
| | | 4.0 | | 478.85±22.27 | **276.64±2.32** | 298.49±3.37 | 446.93±27.17 | 279.83±18.33 | **279.05±2.91** |
| | MinQ | 3.5 | | 376.00±24.65 | 298.31±20.34 | **283.16±4.16** | 423.76±50.93 | 278.75±17.00 | **277.32±17.16** |
| | | 4.0 | | 406.03±6.45 | 315.39±18.80 | **286.90±2.48** | 465.95±46.43 | 297.21±23.12 | **288.77±8.31** |
| $HangZhou_2$ | Gaussian | 3.5 | | 495.92±23.47 | 353.49±6.46 | **343.31±4.65** | 429.53±13.96 | 367.62±8.48 | **352.33±8.51** |
| | | 4.0 | | 520.59±17.34 | 355.93±5.86 | **339.43±3.60** | 432.60±6.00 | 378.09±9.33 | **359.60±11.89** |
| | U-rand | 3.5 | 406.65 | 567.56±20.09 | 415.49±11.33 | **364.17±4.22** | 481.32±37.20 | 426.04±13.45 | **402.33±10.19** |
| | | 4.0 | | 566.64±17.55 | 429.26±11.82 | **370.33±3.10** | 472.05±42.80 | 433.15±9.94 | **397.78±9.88** |
| | MAD | 3.5 | | 496.73±22.83 | **333.93±3.71** | 341.95±6.63 | 433.46±32.89 | 362.01±6.67 | **350.77±9.28** |
| | | 4.0 | | 528.74±28.18 | **339.41±8.00** | 344.82±2.94 | 471.26±29.52 | 363.45±14.04 | 370.91±15.03 |
| | MinQ | 3.5 | | 441.72±31.86 | **345.80±2.82** | 346.51±5.91 | 425.09±30.34 | 356.49±5.33 | 361.75±6.50 |
| | | 4.0 | | 478.9±20.02 | **349.58±7.40** | 350.15±4.85 | 450.80±31.37 | 363.12±6.40 | 369.61±5.97 |

Table 2: **ATT in JiNan and HangZhou: 25% refers to missing data in $sensor_W$, and 50% refers to $sensor_W$ and $sensor_E$.**

| Dataset | Mask Scale | FixedTime base | Advanced-MaxPressure base | RobustLight | RobustLight++ | Advanced-MpLight base | RobustLight | RobustLight++ |
|---|---|---|---|---|---|---|---|---|
| $JiNan_1$ | 25% | 428.11±0.00 | 352.13±0.00 | **296.50±1.12** | 315.25±6.42 | 552.15±120.94 | **371.95±90.21** | 400.35±63.58 |
| | 50% | | 1059.67±0.00 | 610.43±68.52 | **432.15±4.95** | 1045.75±26.83 | 878.09±16.55 | **867.15±60.67** |
| $JiNan_2$ | 25% | 368.76±0.00 | 323.13±0.00 | **273.18±3.95** | 278.87±3.68 | 490.56±92.29 | **276.05±4.71** | 312.99±47.86 |
| | 50% | | 1209.97±0.00 | **755.51±106.57** | 912.18±24.89 | 1082.64±65.15 | 612.14±43.37 | **558.87±71.86** |
| $JiNan_3$ | 25% | 383.01±0.00 | 340.81±0.00 | 281.56±4.11 | **281.05±4.23** | 403.29±30.61 | **288.68±9.41** | 337.98±19.97 |
| | 50% | | 1109.57±0.00 | 570.54±50.20 | **371.44±21.02** | 1061.35±67.51 | 918.43±27.84 | **654.51±83.29** |
| $HangZhou_1$ | 25% | 495.57±0.00 | 530.33±0.00 | 369.52±12.54 | **318.41±5.98** | 478.89±37.35 | 363.55±7.80 | **343.70±10.98** |
| | 50% | | 1186.56±0.00 | 563.56±36.21 | **440.00±94.94** | 867.95±172.63 | 824.83±149.15 | **394.88±43.05** |
| $HangZhou_2$ | 25% | 406.65±0.00 | 409.56±0.00 | 350.86±3.11 | **341.47±1.68** | 373.59±13.70 | 360.22±10.27 | **346.86±6.61** |
| | 50% | | 782.93±0.00 | 447.28±15.87 | **350.47±2.12** | 633.73±89.70 | 459.36±6.87 | **355.15±7.12** |

### 5.2.3 TRANSFER EXPERIMENTS

Table 3 shows that our RobustLight++ generalizes effectively to the unseen SUMO Cologne8 dataset, where the diffusion-based outer-learner is meta-trained on Cityflow datasets Jinan and Hangzhou for zero-shot transfer, the inner-learner is adapted 50 samples on SUMO dataset Cologne8 for few-shot transfer. Both MPLight and MaxPressure get better performance after zero-shot and few-shot meta training in average waiting time (AWT) and ATT of emergency vehicle (EMV) and regular vehicle (REV) used by Su et al. (2022). Under Gaussian and Uniform-Random noise, in most scenarios, the few-shot setting surpasses zero-shot performance for both emergency and regular vehicles, confirming the effectiveness of our method.

Table 3: **Transfer performance (AWT and ATT) of EMV and REV on Cologne8. Results are averaged over 5 runs.**

| Algorithm | Metrics | Noise-Free | Guassian | Zero-shot | Few-shot | U-Rand | Zero-shot | Few-shot |
|---|---|---|---|---|---|---|---|---|
| MPlight | $AWT_{EMV}$ | 8 | 25.0 | 20.0 | **15.0** | 20.0 | 15.0 | **15.0** |
| | $ATT_{EMV}$ | 20 | 25.0 | 20.0 | **15.0** | 20.0 | 15.0 | **15.0** |
| | $AWT_{REV}$ | 0.51 | 2.14 | **1.85** | 1.95 | 2.37 | 2.16 | **2.13** |
| | $ATT_{RMV}$ | 54.40 | 67.47 | 62.50 | **62.17** | 67.49 | 65.57 | **64.96** |
| MaxPressure | $AWT_{EMV}$ | 6.0 | 0.0 | 0.0 | **0.0** | 0.0 | **0.0** | 6.0 |
| | $ATT_{EMV}$ | 25.0 | 15.0 | 15.0 | **15.0** | 15.0 | **15.0** | 20.0 |
| | $AWT_{REV}$ | 0.68 | 2.25 | 1.48 | **1.43** | 3.32 | **2.21** | 2.24 |
| | $ATT_{REV}$ | 56.46 | 66.71 | 61.64 | **61.29** | 72.97 | 65.39 | **64.52** |

Table 4: **Transfer to unseen cities performance of ATT in $JiNan$, $HangZhou$. Results are averaged over 5 runs.**

| Dataset | Noise Type | Noise Scale | FixedTime | Advanced-CoLight | | | | Advanced-MpLight | | | |
|---|---|---|---|---|---|---|---|---|---|---|---|
| | | | | base | RobustLight | RobustLight++ (Zero-Shot) | RobustLight++ (Few-Shot) | base | RobustLight | RobustLight++ (Zero-Shot) | RobustLight++ (Few-Shot) |
| $JiNan_1$ | Gaussian | 3.5 | 428.11 | 316.96 | 423.56 | 329.56 | **285.88** | 327.93 | **322.04** | 380.98 | 369.81 |
| | U-rand | 3.5 | | 483.26 | 540.52 | 419.14 | **360.59** | 417.81 | 328.01 | 340.18 | **325.05** |
| | MAD | 3.5 | | 454.77 | 483.18 | 445.61 | **325.73** | 339.41 | **296.69** | 495.94 | 454.51 |
| | MinQ | 3.5 | | 493.10 | 393.85 | 528.60 | **347.36** | 347.62 | 368.18 | 395.84 | **339.12** |
| | Mask | 25% | | 343.34 | 336.82 | 291.41 | **312.83** | 552.15 | **398.48** | 562.94 | 639.32 |
| $JiNan_2$ | Gaussian | 3.5 | 368.77 | 338.12 | 427.64 | 292.83 | **268.72** | 626.67 | 291.99 | **285.45** | 287.36 |
| | U-rand | 3.5 | | 748.76 | 565.60 | 619.69 | **314.64** | 506.24 | **308.08** | 362.71 | 333.73 |
| | MAD | 3.5 | | 563.99 | 356.23 | 487.48 | **273.49** | 330.25 | 280.68 | 270.75 | **268.21** |
| | MinQ | 3.5 | | 344.12 | 348.00 | 368.26 | **266.12** | 272.16 | 271.29 | 277.54 | **270.85** |
| | Mask | 25% | | 277.85 | 297.95 | 270.94 | **285.39** | 490.56 | **342.39** | 359.96 | 386.45 |
| $JiNan_3$ | Gaussian | 3.5 | 383.01 | 320.04 | 413.53 | 288.25 | **262.37** | 378.13 | 293.93 | 351.54 | **283.70** |
| | U-rand | 3.5 | | 481.55 | 543.19 | 639.80 | **312.58** | 496.63 | **307.64** | 729.41 | 337.57 |
| | MAD | 3.5 | | 446.89 | 401.58 | 335.38 | **260.04** | 442.19 | 272.86 | 268.99 | **263.00** |
| | MinQ | 3.5 | | 376.00 | 364.33 | 318.01 | **262.94** | 423.76 | 268.08 | 370.69 | **261.61** |
| | Mask | 25% | | 324.42 | 309.56 | 304.23 | **287.33** | 403.29 | 360.49 | 588.86 | **335.52** |
| $HangZhou_1$ | Gaussian | 3.5 | 495.57 | 512.63 | 442.47 | 683.91 | **354.15** | 334.03 | 355.64 | 336.00 | **327.65** |
| | U-rand | 3.5 | | 971.03 | 775.62 | 985.47 | **615.19** | 354.63 | 363.64 | 366.68 | **353.87** |
| | MAD | 3.5 | | 751.58 | **513.22** | 860.74 | 719.18 | 308.78 | 547.87 | **312.55** | 318.79 |
| | MinQ | 3.5 | | 506.60 | 467.94 | 678.96 | **369.34** | 320.32 | 507.84 | **306.32** | 317.45 |
| | Mask | 25% | | 418.49 | 334.30 | **324.31** | 375.17 | 478.89 | 442.75 | 456.33 | **371.35** |
| $HangZhou_2$ | Gaussian | 3.5 | 406.65 | 495.92 | 419.85 | 377.31 | **343.34** | 429.53 | 377.48 | 345.56 | **335.54** |
| | U-rand | 3.5 | | 567.56 | 585.79 | 568.48 | **378.70** | 481.32 | **365.99** | 390.19 | 378.26 |
| | MAD | 3.5 | | 496.73 | 481.36 | 369.08 | **357.09** | 433.46 | 492.92 | 322.85 | **321.66** |
| | MinQ | 3.5 | | 441.72 | 464.05 | 368.53 | **344.55** | 425.09 | 444.09 | **317.63** | 328.69 |
| | Mask | 25% | | 348.21 | 359.46 | 374.68 | **340.42** | 373.59 | **347.58** | 372.22 | 361.43 |

Table 4 summarizes the transfer performance of ATT under different noise types across the $JiNan$ and $HangZhou$ datasets. The base results correspond to performance directly evaluated on noisy datasets, while RobustLight and RobustLight++ (Zero-Shot) are trained only on seen cities and transferred to unseen cities (eg. seen city $JiNan$ to unseen $HangZhou$ or seen city $HangZhou$ to $JiNan$), without fine-tuning. RobustLight++ (Few-Shot) further adapts to the target domain using 100 samples. All numbers are averaged over five random seeds. RobustLight++ (Zero-Shot) achieves additional gains and shows statistically significant improvements over RobustLight in most heavy-noise settings (e.g., MAD, Mask), demonstrating stronger cross-city generalization. With only 100 samples, RobustLight++ (Few-Shot) achieves the highest overall performance, often recovering or surpassing clean-data performance. These results highlight the robustness and transferability of RobustLight++, especially under unseen and severe noise.

We also conduct transfer experiments to evaluate the generalization capability of RobustLight++ in more large-scale network like $NewYork$. Specifically, we train models on datasets from $JiNan$ and $HangZhou$, and test them on the $Newyork$ dataset. The Figure 4b demonstrates that RobustLight++ achieves SoTA performance in the transfer setting, further validating the effectiveness and robustness across diverse urban environments, with a 6.92% improvement.

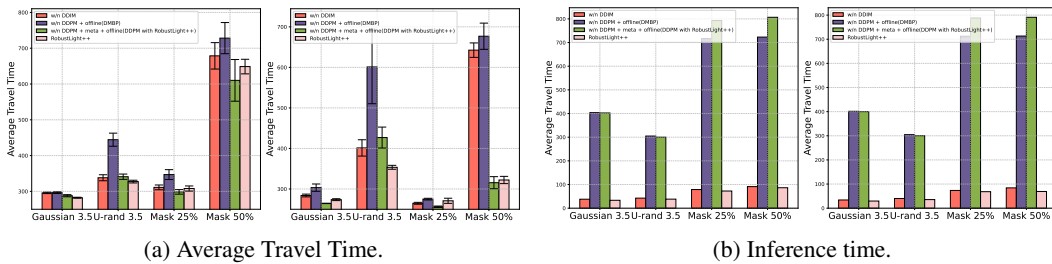

Figure 3: Ablation studies on ATT and inference time in $HangZhou_2$ and $JiNan_2$.

### 5.2.4 ABALATION STUDY

We conduct ablation studies comparing the following configurations: DDIM with online training, DDPM with offline training (DMBP (Yang & Xu, 2023)), RobustLight++ with DDPM, and Robust-Light++. As shown in Figure 3, our proposed method, RobustLight++, achieves SoTA performance. Moreover, it effectively balances performance and inference time, demonstrating practicality.

### 5.2.5 INFERENCE TIME

Compared to RobustLight, our method uses the DDIM sampling strategy to significantly speed up both denoising and demasking. As shown in Table 17, it reduces runtime by 87.9% for denoising and 91.9% for demasking, leading to much greater inference efficiency and making the framework more practical for real-time deployment in large-scale traffic networks.

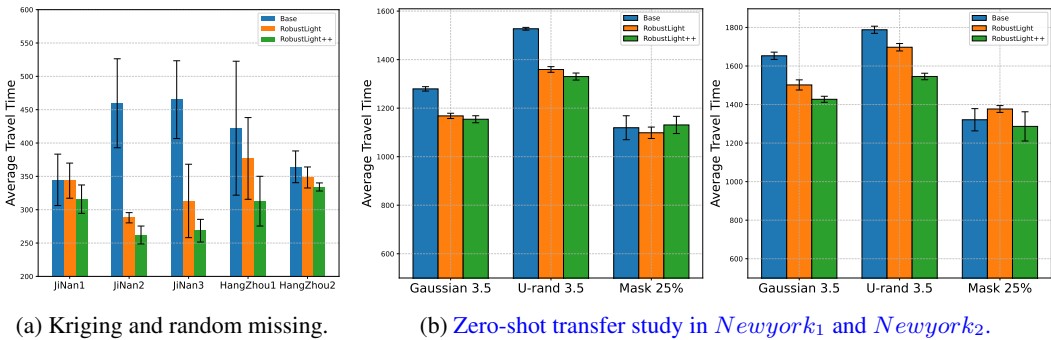

(a) Kriging and random missing.  (b) Zero-shot transfer study in $Newyork_1$ and $Newyork_2$.

Figure 4: Transfer and other experiments study.

### 5.2.6 KRIGING AND MISSING EXPERIMENTS

To validate our approach in multiple missing types, we set up involved random masking of data from Kriging Missing (12.5%, single-intersection-sensor failure) and Random Missing (12.5%, full-intersection failure). As demonstrated in Figure 4a, our method effectively addresses data missing scenarios and exhibits robust performance , with an average improvement of 10.57%.

## 6 CONCLUSION

In this paper, we proposed a diffusion-based TSC framework that addresses key limitations of prior methods like RobustLight. Unlike previous approaches that required separate models per-city and struggled with generalization, our method introduced a meta-learnable DDIM-based controller enabling robust cross-city adaptation and efficient inference. Experiments showed that our approach improved control performance in unseen cities and reduced denoising and demasking times by 87.9% and 91.9%, respectively. These results show the framework's potential for scalable real-time use in complex urban networks. Future work will explore real-world deployment feedback and extend the method to multi-agent coordination with limited communication.

ETHICS STATEMENT

This work adheres to the ICLR Code of Ethics. Our research does not involve human subjects or sensitive data, and we have ensured compliance with all relevant legal and ethical standards. The proposed methodology, which focuses on [briefly describe your work, e.g., novel neural network architectures], does not introduce harmful applications or exacerbate bias beyond existing benchmarks. To address potential fairness concerns, we evaluated our model across diverse datasets, as detailed in appendix, ensuring robustness and fairness in performance. Any potential conflicts of interest, including funding sources, are disclosed in the acknowledgments section. We have strived to maintain research integrity by providing clear documentation and transparent evaluation metrics throughout the paper and supplementary materials.

REPRODUCIBILITY STATEMENT

To ensure the reproducibility of our results, we have provided comprehensive details in the main paper and supplementary materials. The source code for our proposed model and experiments is available anonymously at `https://anonymous.4open.science/r/RobustLightPlus-E14F.`. All datasets used are publicly available, with preprocessing steps fully documented in the appendix. For theoretical contributions, we include complete proofs of our claims in main text. Hyperparameters, training procedures, and evaluation metrics are detailed in appendix and the supplementary materials to facilitate replication of our experiments.

LLM USAGE STATEMENT

In the preparation of this manuscript, LLM was used solely for polishing the text to improve clarity, grammar, and readability. The LLM did not contribute to the research ideation, methodology, analysis, or core writing of the paper. All scientific content, including ideas, experiments, and results, was developed and authored by the human researchers. We take full responsibility for the content of this paper, including the polished text, and confirm that the use of the LLM does not constitute plagiarism or scientific misconduct.

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

## A  DETAILED ALGORITHMS

The Cross-City Diffusion Meta-Learning algorithm, outlined in Algorithm 1, is designed to enable effective knowledge transfer across multiple city-specific tasks for diffusion-based models. By leveraging a meta-learning framework inspired by Reptile, the algorithm initializes a shared parameter set $\theta$ and iteratively updates it using gradients computed from source tasks $\mathcal{T}_i^{\text{src}}$. For each source task, it performs an inner update with step size $\eta$ to compute task-specific parameters $\theta_i'$, aggregates the parameter differences, and updates the global parameters using a meta-learning rate $\mu$. After training on source tasks, the model is fine-tuned on the target task $\mathcal{T}_{\text{tgt}}$ to adapt to specific city characteristics, enhancing generalization in diffusion-based applications such as urban data modeling.

The Repaint Algorithm of RobustLight++, presented in Algorithm 2, is a robust iterative method for reconstructing signals in diffusion-based models, particularly suited for tasks requiring inpainting or signal recovery under noisy conditions. The algorithm takes as input an initial signal estimate $\tilde{s}_t^i$, a context $c_{t-1}$, and a mask $m$, and performs $K$ iterations, each with $U$ inner steps. In each step, it samples noise $\epsilon$ and $z$ from a standard normal distribution (except in the first iteration, where noise is set to zero) and applies Equations (10), (11), and (12) to update known and unknown signal components and recover the signal. A stochastic update is applied when necessary, governed by the parameter $\beta_{i-1}$, to introduce controlled noise, ultimately producing a refined signal estimate $\hat{s}_t$ that enhances robustness in applications like image or data reconstruction.

---

**Algorithm 1** Cross-City Diffusion Meta-Learning

---

1: **Input:** Source tasks $\{\mathcal{T}_i^{\mathrm{src}}\}$, target $\mathcal{T}_{\mathrm{tgt}}$, step sizes $\alpha, \eta$
2: Initialize $\theta$
3: **for** each outer iteration **do**
4:     $\phi \leftarrow 0$
5:     **for** each source task $\mathcal{T}_i^{\mathrm{src}}$ **do**
6:         Compute $\phi \leftarrow \theta - \eta \, \nabla_\theta L_{\mathrm{diff}}(\theta; \mathcal{T}_i)$
7:         $\hat{\theta} \leftarrow \hat{\theta} + (\phi - \theta_i)$
8:     **end for**
9:     $\theta \leftarrow \theta + \frac{\mu}{N} \sum_{i=1}^{N} \hat{\theta}$
10: **end for**
11: Few-shot training $\theta$ on $\mathcal{T}_{\mathrm{tgt}}$ via $L_{\mathrm{diff}}$
12: **return** $\theta$

---

**Algorithm 2** Repaint algorithm of RobustLight++

---

1: Input $\tilde{s}_t^j, c_{t-1}, m$
2: **for** $j = 1$ to $K$ **do**
3:     **for** $u = 1$ to $U$ **do**
4:         $\epsilon \sim \mathcal{N}(0, I)$ if $j > 1$, else $\epsilon = 0$
5:         Get $\tilde{s}_{t,known}^{j-1}$ by Equation (10)
6:         $z \sim \mathcal{N}(0, I)$ if $j > 1$, else $z = 0$
7:         Get $\tilde{s}_{t,unknown}^{j-1}$ by Equation (11)
8:         Get recovered $\tilde{s}_t^{j-1}$ by Equation (12)
9:         **if** $u < U$ and $j > 1$ **then**
10:             $\tilde{s}_t^j \sim \mathcal{N}(\sqrt{1 - \beta_{j-1}}\tilde{s}_t^{j-1}, \beta_{j-1}I)$
11:         **end if**
12:     **end for**
13: **end for**
14: Return $\hat{s}_t$

---

## B   PROOFS OF THEOREM 1

*Proof of Theorem 1.* Recall the deterministic probability flow ODE corresponds to DDIM by Song et al. (2020b). The diffusion step can be written:

$$\frac{d\mathbf{x}}{dt} = \mathbf{f}(\mathbf{x}, t) - \frac{1}{2} g(t)^2 \nabla \log p_t(\mathbf{x}). \tag{15}$$

In the DDIM-Reptile framework, the model approximates this field using a learned score network $\mathbf{s}_\theta(\mathbf{x}, t) \approx \nabla \log p_t(\mathbf{x})$. The model's approximate velocity field is:

$$v(x, t) = \frac{d\mathbf{x}}{dt} \approx \mathbf{f}(\mathbf{x}, t) - \frac{1}{2} g(t)^2 \mathbf{s}_\theta(\mathbf{x}, t). \tag{16}$$

Let $q_t$ denote the distribution of $x(t)$ under this approximate ODE (with initial distribution $q_T$ at $t = T$). By construction $p_T$ and $q_T$ are the terminal distributions for the true and model processes; if the model precisely matches the chosen noise prior. At time $t = 0$, we have $p_0$ (true data) and $q_0$ (model-generated data). Our goal is to bound $W_2(p_0, q_0)$. For each $t \in [0, T]$, let $p_t$ denote the noised data distribution and $q_t$ the corresponding model distribution at time $t$. The score function of $p_t$ is

$$s^\star(x, t) := \nabla_x \log p_t(x). \tag{17}$$

$s_\theta(x, t)$ is the trained score network, and define the score matching loss as

$$L_{\mathrm{diff}}(\theta) := \frac{1}{2} \int_0^T \lambda(t) \, \mathbb{E}_{x \sim p_t} \left[ \|\nabla_x \log p_t(x) - s^\star(x, t)\|^2 \right] dt, \tag{18}$$

for some positive weighting function $\lambda(t) > 0$. According to Song et al. (2020b), minimizing Eq. 8 corresponds to estimating the true score function $s^\star(x, t) = \nabla_x \log p_t(x)$ in the $L^2$ sense.

From Kwon et al. (2022) Theorem 1, using Cauchy–Schwarz on that time-integral, we obtain an upper bound in terms of the root of the integrated mean-square error of the score:

$$W_2(p_0, q_0^{\text{cont}}) \leq \sqrt{2 \int_0^T g(t)^4 I(t)^2 \, dt \cdot \int_0^T \lambda(t) \mathbb{E}_{p_t} \left[ \|\nabla \log p_t(x) - s_\theta(x,t)\|^2 \right] dt} \tag{19}$$
$$+ I(T) W_2(p_T, q_T),$$

The first factor $\int_0^T g(t)^4 I(t)^2 dt$ is a constant determined by the diffusion schedule $g(t)$ and the integrated Lipschitz coefficients $I(t)$. We may therefore write

$$W_2(p_0, q_0^{\text{cont}}) \leq \mathcal{C}_{\text{score}} \sqrt{L_{\text{diff}}(\theta)} + \mathcal{C}_{\text{init}} W_2(p_T, q_T), \tag{20}$$

for some constant $\mathcal{C}_{\text{score}} = \sqrt{2 \int_0^T g(t)^4 I(t)^2 dt}$ and $\mathcal{C}_{\text{init}} = I(T)$.

In practice, DDIM generation runs the ODE backward in $N$ discrete steps (from $T$ to $0$). Let the time step be $\Delta t = T/N$, and denote $\mu = \Delta t$ for notational consistency with the theorem. The Euler discretization of the reverse dynamics is

$$x_{t-\Delta t} = x_t + \Delta t \Big( f(x_t, t) - \frac{1}{2} g(t)^2 s_\theta(x_t, t) \Big), \tag{21}$$

This update mirrors the Reptile meta-learning rule $\theta \leftarrow \theta + \mu(\phi - \theta)$, where $\mu$ is the meta step size. Here, each diffusion step plays the role of one Reptile meta-update, and $\mu = \Delta t$ controls the discretization resolution.

Let $q_0^{\text{cont}}$ denote the distribution induced by the continuous-time ODE, and $q_0^{\text{disc}}$ the distribution obtained by the Euler discretization. Assume the vector field $v(x,t)$ satisfies the one-sided Lipschitz assumption proposed by Donchev & Farkhi (1998) (Definition 2.1) with coefficient $L_s(t)$:

$$\langle v(x,t) - v(y,t), \, x - y \rangle \leq L_s(t) \|x - y\|^2, \qquad \forall x, y \in \mathbb{R}^d, \ t \in [0, T], \tag{22}$$

By Theorem 3.2 and 4.3 of Donchev & Farkhi (1998), we get the pathwise error bound

$$\sup_{0 \leq t \leq T} \|x_{\text{cont}}(t) - x_{\text{disc}}(t)\| \leq C_{\text{disc}} \mu, \tag{23}$$

Let $q_0^{\text{cont}}$ and $q_0^{\text{disc}}$ denote the laws of the continuous and Euler-approximated reverse flows at time $t = 0$. According to the definition of 2-Wasserstein or Monge-Kantorovich distance Kwon et al. (2022) and combined with Eq. 23, we get:

$$W_2\big(q_0^{\text{cont}}, q_0^{\text{disc}}\big) \leq \big(\mathbb{E}\|X_0^{\text{cont}} - X_0^{\text{disc}}\|^2\big)^{1/2} \leq C_{\text{disc}} \, \mu, \tag{24}$$

From Theorem 4.3 of Donchev & Farkhi (1998), we get the discretization constant admits the explicit form

$$C_{\text{disc}} = c_0(C_\tau + C_\chi + 1), \qquad c_0 = \exp\Big( \int_0^T L_s^+(t) \, dt \Big) \max\{2, B\}, \tag{25}$$

In particular, the one-sided Lipschitz property prevents exponential error amplification in the reverse diffusion ODE, ensuring that discretization error accumulates only linearly with $\mu$.

Finally, let $q_0 \equiv q_0^{\text{disc}}$ be the practical model distribution. By the triangle inequality,

$$W_2(p_0, q_0) \leq W_2(p_0, q_0^{\text{cont}}) + W_2(q_0^{\text{cont}}, q_0^{\text{disc}}), \tag{26}$$

Substituting the continuous-time bound (from the score approximation) and the discretization bound equation 24, we obtain

$$W_2(p_0, q_0) \leq \mathcal{C}_{\text{score}} \sqrt{L_{\text{diff}}(\theta)} + \mathcal{C}_{\text{disc}} \cdot \mu + \mathcal{C}_{\text{init}} W_2(p_T, q_T). \tag{27}$$

which completes the proof.

## C  DATASETS AND COMPARED METHODS

### C.1  DATASETS

We use real-world traffic flow and road topology datasets for our experiments, with Cityflow Zhang et al. (2019) as the simulator to evaluate ATT and exit points with a simulation time of 60 minutes for all vehicles. The datasets include vehicle start and end points, following a fixed motion model. Seven traffic datasets from three cities, JiNan and HangZhou (China), and New York Zheng et al. (2019) (USA, are used.

- **JiNan Datasets:** The JiNan road network consists of 12 intersections (in a $3 \times 4$ grid). It includes three traffic flow datasets: $JiNan_1$, $JiNan_2$, and $JiNan_3$.
- **HangZhou Datasets:** The HangZhou network encompasses 16 intersections (in a $4 \times 4$ grid) and features two datasets: $HangZhou_1$ and $HangZhou_2$.
- **New York Datasets:** The New York network features a more complex structure with 192 intersections ($28 \times 7$ grid) and includes two datasets: $Newyork_1$ and $Newyork_2$.

### C.2  COMPARED METHODS

#### C.2.1  TRADITIONAL METHODS

These methods include FixedTime Webster (1958), which uses a fixed green phase time; Advanced-Maxpressure Zhang et al. (2022), which uses running and waiting vehicles to choose the phase; and Maxpressure Gershenson (2004), which uses waiting vehicles to choose the phase.

#### C.2.2  RL-BASED METHODS

For RL benchmarks, we consider CoLight Wei et al. (2019c), which uses waiting and neighboring vehicles to select the signal phase; Advanced-CoLight Zhang et al. (2022), which employs waiting and running vehicles with a graph attention neural network; and Advanced-Mplight Zhang et al. (2022), which uses the FRAP Zheng et al. (2019) model for signal phase selection.

#### C.2.3  ROBUSTLIGHT METHOD

RobustLight and RobustLight++ integrate the base methods of traditional and RL-based TSC algorithms to recover data in real-time, and then evaluates the ATT under different sensor noise attacks and sensor damage. Results are presented as the average of ten independent runs.

## D  SENSOR NOISE AND SENSOR DAMAGE EXPAND EXPERIMENTS

Table 5: **Performance of ATT in** $JiNan_2$**,** $HangZhou_1$**. Our RobustLight++ recovers the state of traditional and RL-based TSC algorithms.**

| Dataset | Noise Type | Noise Scale | FixedTime base | Advanced-CoLight base | RobustLight | RobustLight++ | Advanced-MpLight base | RobustLight | RobustLight++ | Advanced-MaxPressure base | RobustLight | RobustLight++ |
|---|---|---|---|---|---|---|---|---|---|---|---|---|
| $JiNan_2$ | Gaussian | 3.5 | | 338.12±12.82 | 292.70±2.31 | **273.68±2.32** | 626.67±177.54 | 275.78±2.17 | **268.58±1.28** | 276.10±1.50 | 280.43±1.77 | **269.98±0.81** |
| | | 4.0 | | 357.56±30.63 | 308.17±3.60 | **265.32±1.31** | 689.61±202.25 | 284.35±2.65 | **273.24±4.24** | 281.06±1.67 | 287.71±3.12 | **274.22±1.07** |
| | U-rand | 3.5 | 368.77 | 748.76±97.15 | 564.09±61.36 | **353.52±4.84** | 506.24±105.45 | 297.90±4.67 | **290.15±7.68** | 301.54±2.38 | **300.41±1.79** | 304.85±4.04 |
| | | 4.0 | | 784.89±68.08 | 611.86±54.28 | **334.67±4.53** | 572.98±126.74 | 303.68±5.53 | **295.55±8.74** | 307.22±0.96 | **307.13±1.4** | 320.89±3.23 |
| | MAD | 3.5 | | 563.99±40.13 | **273.66±4.84** | 275.88±1.76 | 330.25±130.61 | **254.85±2.30** | 262.66±0.89 | - | - | - |
| | | 4.0 | | 630.31±19.84 | **277.99±1.23** | 286.36±2.13 | 454.53±256.35 | **257.66±3.72** | 279.84±0.91 | - | - | - |
| | MinQ | 3.5 | | 344.12±29.23 | **287.39±12.74** | 292.35±8.27 | 272.16±8.66 | **256.80±2.05** | 263.82±1.75 | - | - | - |
| | | 4.0 | | 367.51±55.65 | 295.08±16.33 | **288.83±4.36** | 402.47±141.56 | **261.35±3.43** | 277.89±2.16 | - | - | - |
| $HangZhou_1$ | Gaussian | 3.5 | | 512.63±25.89 | 363.83±8.25 | **337.84±2.34** | 334.03±2.48 | 351.44±9.07 | **315.75±1.77** | 327.94±1.03 | 346.14±1.58 | **319.10±0.83** |
| | | 4.0 | | 530.59±30.78 | 388.49±7.73 | **336.53±2.89** | 338.76±3.89 | 381.53±13.58 | **316.92±1.54** | 331.25±1.28 | 370.61±2.56 | **320.27±0.43** |
| | U-rand | 3.5 | 495.57 | 971.03±34.19 | 537.52±56.18 | **426.79±23.70** | 354.63±2.05 | **332.80±1.60** | 343.87±5.61 | 355.51±3.32 | **348.57±1.07** | 350.61±1.76 |
| | | 4.0 | | 982.41±42.18 | 561.60±50.29 | **408.59±14.11** | 360.11±2.75 | **334.80±1.67** | 351.45±3.08 | 360.88±3.08 | **353.11±1.71** | 363.21±4.21 |
| | MAD | 3.5 | | 751.58±44.00 | **319.42±1.81** | 333.67±3.23 | 308.78±3.91 | 312.76±4.85 | **309.87±2.10** | - | - | - |
| | | 4.0 | | 754.66±37.04 | **323.48±2.45** | 352.27±4.76 | 313.53±3.06 | **318.14±5.43** | 331.27±4.13 | - | - | - |
| | MinQ | 3.5 | | 506.60±42.19 | **35.29±3.50** | 351.39±11.40 | 320.32±6.34 | **314.07±4.20** | 315.70±3.19 | - | - | - |
| | | 4.0 | | 550.00±37.44 | **343.82±5.70** | 363.18±14.05 | 324.65±5.88 | **323.04±4.15** | 330.05±5.85 | - | - | - |

Table 6: **ATT in JiNan and HangZhou: 25% refers to missing data in** $sensor_W$**, and 50% refers to** $sensor_W$ **and** $sensor_E$**.**

| Dataset | Mask Scale | Advanced-CoLight | | |
| --- | --- | --- | --- | --- |
| | | base | RobustLight | RobustLight++ |
| JiNan1 | 25% | 343.34±8.39 | 310.14±9.63 | **307.98±7.15** |
| | 50% | 699.02±35.07 | **539.30±54.88** | 648.73±20.38 |
| JiNan2 | 25% | 277.85±7.50 | **266.62±3.79** | 270.90±6.33 |
| | 50% | 682.61±29.58 | 351.08±17.45 | **322.20±8.94** |
| JiNan3 | 25% | 324.42±13.55 | 278.06±9.84 | **267.95±6.13** |
| | 50% | 627.95±53.19 | 417.49±111.67 | **306.57±7.86** |
| HangZhou1 | 25% | 418.49±13.59 | **331.65±7.68** | 383.35±12.47 |
| | 50% | 624.83±13.27 | 434.31±34.36 | **415.23±7.23** |
| HangZhou2 | 25% | 348.21±7.21 | 340.80±4.64 | **335.02±2.35** |
| | 50% | 499.34±28.56 | 453.90±26.08 | **352.05±6.22** |

Table 7: **Performance of ATT in** $JiNan$**,** $HangZhou$**. Our RobustLight++ recovers the state of traditional and RL-based TSC algorithms to evaluate the performance.**

| Dataset | Noise Type | Noise Scale | Advanced-MaxPressure | | |
| --- | --- | --- | --- | --- | --- |
| | | | base | RobustLight | RobustLight++ |
| $JiNan_1$ | Gaussian | 3.5 | 285.72±2.20 | 283.47±1.16 | **279.85±1.20** |
| | | 4.0 | 289.74±2.61 | 288.90±1.18 | **283.95±1.30** |
| | U-rand | 3.5 | 312.67±2.81 | **307.08±2.67** | 307.55±2.07 |
| | | 4.0 | 321.26±1.81 | **312.52±2.82** | 334.73±1.70 |
| $JiNan_3$ | Gaussian | 3.5 | 268.67±1.56 | 268.69±1.32 | **262.13±0.60** |
| | | 4.0 | 273.22±0.77 | 272.95±1.68 | **265.42±1.56** |
| | U-rand | 3.5 | 293.44±2.74 | 291.58±1.74 | **285.97±1.11** |
| | | 4.0 | 300.53±3.67 | 296.06±2.23 | **295.45±1.85** |
| $HangZhou_2$ | Gaussian | 3.5 | 345.87±1.37 | 346.53±1.61 | **341.09±1.08** |
| | | 4.0 | 348.39±2.53 | 350.68±1.44 | **345.84±1.80** |
| | U-rand | 3.5 | 362.58±0.93 | **359.44±1.02** | 359.94±0.64 |
| | | 4.0 | 366.46±2.44 | **363.44±1.91** | 374.26±2.71 |

The Table 6 presents the performance of the Advanced-CoLight algorithm across five datasets under two data missing scenarios: 25% missing data in $sensor_W$ and 50% missing data in both $sensor_W$ and $sensor_E$. It compares the base model, RobustLight, and RobustLight++, highlighting that RobustLight++ generally achieves the best or competitive performance, with notable improvements in heavily degraded conditions (e.g., 50% missing data), as indicated by the highlighted cells showing lower mean values and reduced standard deviations.

The Table 7 evaluates the Advanced-MaxPressure algorithm's performance across three datasets ($JiNan_1$, $JiNan_3$, $HangZhou_2$) under various conditions, likely involving noise or missing data, with four entries per dataset. RobustLight++ consistently outperforms or matches the base and RobustLight models, as evidenced by the highlighted cells with lower mean values and smaller standard deviations, demonstrating its effectiveness in recovering and stabilizing the performance of traditional and reinforcement learning-based traffic signal control algorithms.

The Table 5 presents the performance of the Advanced-CoLight, Advanced-MpLight and Advanced-MaxPressure algorithms, evaluated on two distinct datasets: $JiNan_2$ and $HangZhou_1$. The experi-

ments were conducted under eight different scenarios for each dataset, designed to simulate environments with data imperfections like noise. The empirical results reveal that the RobustLight++ framework outperforms both the baseline model and the RobustLight implementation in many scenarios. This superiority is quantitatively evidenced by the consistently lower mean values and smaller standard deviations highlighted in the table, which respectively indicate higher operational efficiency and greater stability.

# E ABALATION EXPAND EXPERIMENTS

Table 8: **ATT ablation of RobustLight++ based on Advanced-Colight in $JiNan_2$ and $HangZhou_2$ under noise and mask.**

| Dataset | Type | Scale | Advanced-CoLight | | | |
|---|---|---|---|---|---|---|
| | | | w/n ddim | w/n ddpm+offline | w/n ddpm+meta+offline | RobustLight++ |
| $JiNan_2$ | Gaussian | 3.5 | 283.96±3.42 | 285.38±4.03 | **264.37±0.40** | 273.68±2.32 |
| | U-rand | 3.5 | 401.34±20.15 | 601.13±90.80 | 426.87±25.85 | **353.52±4.84** |
| | Mask | 25% | 264.73±2.33 | 256.35±2.49 | **256.21±1.74** | 270.90±6.33 |
| | Mask | 50% | 642.50±17.84 | **283.83±9.48** | 315.55±14.91 | 322.20±8.94 |
| $HangZhou_2$ | Gaussian | 3.5 | 340.44±4.40 | 360.36±13.92 | **339.82±2.74** | 343.31±4.65 |
| | U-rand | 3.5 | 364.90±1.64 | 573.24±17.7 | 441.79±21.83 | **364.17±4.22** |
| | Mask | 25% | 345.65±1.72 | 348.35±11.41 | 345.71±6.14 | **335.02±2.35** |
| | Mask | 50% | 485.30±10.76 | 381.7±12.33 | **349.10±6.76** | 352.05±6.22 |

Table 9: **Inference time ablation of RobustLight++ based on Advanced-Colight in $JiNan_2$ and $HangZhou_2$ under noise and mask.**

| Dataset | Type | Scale | Advanced-CoLight | | | |
|---|---|---|---|---|---|---|
| | | | w/n ddim | w/n ddpm+offline | w/n ddpm+meta+offline | RobustLight++ |
| $JiNan_2$ | Gaussian | 3.5 | 34.02 | 409.08 | 407.72 | **29.61** |
| | U-rand | 3.5 | 40.17 | 328.03 | 389.25 | **35.61** |
| | Mask | 25% | 73.50 | 710.01 | 766.5 | **68.41** |
| | Mask | 50% | 84.14 | 716.6 | 780.24 | **69.27** |
| $HangZhou_2$ | Gaussian | 3.5 | 48.99 | 417.00 | 415.53 | **40.06** |
| | U-rand | 3.5 | 53.31 | 333.5 | 328.85 | **43.55** |
| | Mask | 25% | 97.77 | 705.93 | 761.54 | **80.51** |
| | Mask | 50% | 93.79 | 706.78 | 764.46 | **81.66** |

Table 10: Ablation study few-shot (100 samples) transfer performance from $JiNan$ to $HangZhou$ under Different Noise Types and Scales based on Advanced-Color-Light in different learning rates and diffusion steps.

| Dataset | Noise Type | Scale | Base | RobustLight++ ($\mu$=0.1,T=100) | RobustLight++ ($\mu$=0.05,T=100) | RobustLight++ ($\mu$=0.05,T=50) |
|---|---|---|---|---|---|---|
| $HangZhou_1$ | Gaussian | 3.5 | 512.63 | 354.15 | **324.87** | 564.89 |
| | U-rand | 3.5 | 971.03 | **615.19** | 686.54 | 798.36 |
| | MinQ | 3.5 | 751.58 | 719.18 | **393.27** | 791.02 |
| | MAD | 3.5 | 506.60 | **369.34** | 372.98 | 381.22 |
| | Mask | 25% | 418.49 | **375.17** | 392.13 | 497.86 |
| $HangZhou_2$ | Gaussian | 3.5 | 495.92 | 343.34 | **342.68** | 361.82 |
| | U-rand | 3.5 | 567.56 | **378.70** | 471.45 | 511.46 |
| | MinQ | 3.5 | 496.73 | **357.09** | 373.30 | 379.58 |
| | MAD | 3.5 | 441.72 | **344.55** | 373.49 | 376.59 |
| | Mask | 25% | 348.21 | **340.42** | 343.54 | 355.51 |

Tables 8 and 9 present the performance and inference time comparisons of RobustLight++ under various noise types and mask conditions, based on ablation studies with the Advanced-CoLight framework. Table 8 shows the ATT results on two datasets ($JiNan_2$ and $HangZhou_2$) with different perturbation types (Gaussian, U-rand) and mask levels (25%, 50%). The comparison includes different model settings: using only DDIM, using DDPM+offline, and using DDPM+meta+offline. RobustLight++ consistently achieves the lowest ATT across most scenarios, demonstrating superior

robustness and performance under noise and partial observability. Table 9 reports the inference time under the same settings. RobustLight++ shows significantly lower inference times compared to all other variants, highlighting its practical efficiency and suitability for real-time deployment. Table 10 shows the few-shot transfer results from $JiNan$ to $HangZhou$ under different noise settings. RobustLight++ reports recovered performance under different learning rates and diffusion steps with few-shot adaptation (100 samples). The setting meta learning rate is 0.1 and diffusion steps is 100 yields the best overall performance.

## F    KRIGING AND RANDOM MISSING EXPAND EXPERIMENTS

Table 11: **ATT in JiNan and HangZhou with 12.5% random and kriging missing data.**

| Dataset | Advanced-CoLight | | |
| --- | --- | --- | --- |
| | base | RobustLight | RobustLight++ |
| $JiNan_1$ | 344.78±38.56 | 343.56±26.34 | **315.84±21.27** |
| $JiNan_2$ | 459.64±66.70 | 287.95±7.69 | **262.11±13.41** |
| $JiNan_3$ | 465.06±58.32 | 313.15±55.05 | **268.51±17.03** |
| $HangZhou_1$ | 422.23±100.44 | 376.90±61.27 | **312.82±37.3** |
| $HangZhou_2$ | 364.32±23.81 | 348.33±15.77 | **334.11±6.07** |

The Table 11 evaluates the performance of the Advanced-CoLight algorithm across five datasets under a 12.5% random and kriging missing data scenario. It compares the base model, Robust-Light, and RobustLight++, with RobustLight++ consistently achieving the lowest mean values (highlighted in gray), indicating superior performance and stability, particularly in $JiNan_2$ (262.11) and $JiNan_3$ (268.51), where it significantly outperforms the base and RobustLight models.

## G    TRANSFER EXPAND EXPERIMENTS

Table 12: **Performance of RobustLight++ based on Advanced-Colight in $Newyork$ transfer by $JiNan_1$**

| Dataset | Type | Scale | Advanced-CoLight | | |
| --- | --- | --- | --- | --- | --- |
| | | | base | RobustLight | RobustLight++ |
| $Newyork_1$ | Gaussian | 3.5 | 1279.46±9.07 | 1168.11±10.90 | **1154.30±14.42** |
| | U-rand | 3.5 | 1527.23±5.64 | 1359.38±11.71 | **1330.24±14.54** |
| | Mask | 25% | 1119.34±49.36 | **1098.55±23.48** | 1130.83±35.49 |
| $Newyork_2$ | Gaussian | 3.5 | 1653.00±18.94 | 1501.80±26.33 | **1427.23±15.71** |
| | U-rand | 3.5 | 1788.00±18.51 | 1697.33±19.22 | **1545.65±16.86** |
| | Mask | 25% | 1321.09±57.72 | 1376.90±17.63 | **1286.63±75.82** |

The Table 12and Figure 5 assess the Advanced-CoLight algorithm's performance across six datasets ($JiNan_2$, $JiNan_3$, $HangZhou_1$, $HangZhou_2$, $Newyork_1$, $Newyork_2$) under various noise conditions (Gaussian at 3.5, U-rand at 3.5, and Mask at 25%), transferred from inner learner of $JiNan_1$ and outer learner of $JiNan_2$, $JiNan_3$, $HangZhou_1$, $HangZhou_2$. RobustLight++ demonstrates enhanced performance with lower mean values in most cases (highlighted in gray), such as 1154.30 for $Newyork_1$ Gaussian and 1427.23 for $Newyork_2$ Gaussian, though it underperforms in the Mask scenario for $Newyork_1$ (1130.83), suggesting its effectiveness varies with noise type.

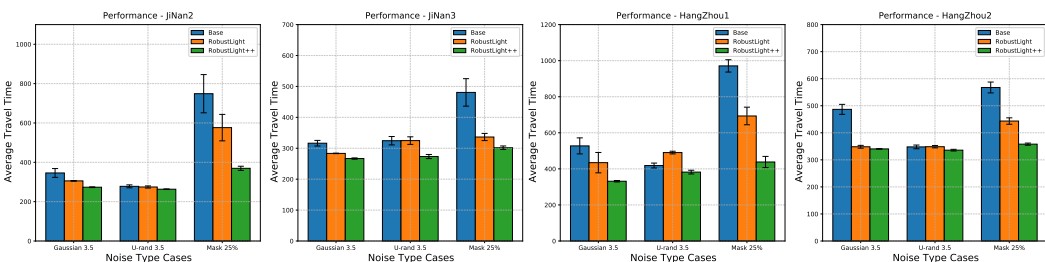

Figure 5: **Performance of RobustLight++ based on Advanced-Colight in** $JiNan_2$, $JiNan_3$, $HangZhou_1$ **and** $HangZhou_2$ **transfer by** $JiNan_1$

Table 13: Training-Time Complexity under Different Meta-Diffusion Settings

| Scenario | #Tasks | Diffusion Steps | Meta Iter. | Inner Steps | Wall-clock (s) | Avg/Step (s) | Peak GPU (MiB) |
|---|---|---|---|---|---|---|---|
| 2tasks_T50 | 2 | 50 | 3 | 30 | 31.93 | 10.64 | 163.06 |
| 3tasks_T50 | 3 | 50 | 3 | 30 | 42.15 | 14.05 | 163.06 |
| 3tasks_T100 | 3 | 100 | 3 | 30 | 42.75 | 14.25 | 163.06 |
| 5tasks_T150 | 5 | 150 | 3 | 30 | 62.57 | 20.86 | 163.06 |

## H TRAINING COMPLEXITY ANALYSIS

We evaluate the training-time complexity of RobustLight++ under various meta-learning and diffusion configurations, as summarized in Table 13. During training, each scenario involves a set of outer tasks corresponding to different city environments. For each meta-iteration, the diffusion model performs $T$ denoising steps, followed by $K$ inner-loop gradient updates for control policy adaptation. Specifically, we fix the number of meta-iterations to 3 and inner training steps to 30, and vary the number of tasks and diffusion depth (50–150 steps) to evaluate scalability.

All experiments are conducted on a single RTX 4090 GPU. Importantly, the peak GPU memory remains stable across all settings, demonstrating that the diffusion-based outer-learner does not introduce excessive memory overhead even with increased task scale or diffusion depth. The results indicate that wall-clock time grows approximately linearly with the number of cities and diffusion steps, while the average inner loss consistently decreases as task diversity increases, validating the effectiveness of meta-training across heterogeneous urban environments.

## I ADDITIONAL EXPERIMENTS

The tables 15 and 16 compare the performance of RobustLight++ against RobustLight across datasets $JiNan_1$, $JiNan_2$, and $HangZhou_2$. For PSNR (higher is better, shown in Table 15), RobustLight++ consistently outperforms others, with notable improvements to 20.87 for $JiNan_1$ Gaussian and 4.17 for $HangZhou_2$ at 50% mask. For MAE (lower is better, shown in Table 16), RobustLight++ also excels, achieving 2.35 for $JiNan_1$ Gaussian and 0.99 for $HangZhou_2$ at 50% mask, demonstrating superior robustness and accuracy across various scales and noise types.

As shown in Table 14, RobustLight++ based on $\pi$-Light Gu et al. (2024) consistently achieves superior performance across both datasets and noise settings. Under both Gaussian and uniform-random perturbations, RobustLight++ yields the lowest values in most evaluation metrics, indicating improved robustness against noisy observations.

## J HYPERPARAMETERS

The Table 18 outlines the hyperparameters used for training the proposed model, categorized into UNet/Model Hyperparameters, Diffusion Training Hyperparameters, and TSC RL Agent Training Hyperparameters. Key settings include an embedding dimension of 64 and a hidden dimension of 256 for the UNet model, with state and action dimensions tailored to 20/32 and 4, respectively. Diffusion training employs a non-Markovian step of 6, a beta schedule of [3.0651, 24.552, -3.1702], a diffusion timestep of 100, and an Adam optimizer with a learning rate of 0.0003, alongside meta

Table 14: Performance on $JiNan_3$ and $HangZhou_1$ under Different Noise Types and Scales based on $\pi$-Light.

| Dataset | Noise Type | Scale | Base | RobustLight | RobustLight++ |
|---------|-----------|-------|------|-------------|---------------|
| $JiNan_3$ | Gaussian | 3.5 | 472.80±45.14 | 441.67±69.53 | **382.73±8.35** |
| | U-rand | 3.5 | 452.44±28.29 | 515.18±16.18 | **366.31±6.99** |
| | MinQ | 3.5 | – | – | – |
| | MAD | 3.5 | – | – | – |
| | Gaussian | 4.0 | 466.44±25.91 | 418.19±58.81 | **387.13±21.41** |
| | U-rand | 4.0 | 457.13±29.63 | 522.79±31.16 | **370.44±13.21** |
| | MinQ | 4.0 | – | – | – |
| | MAD | 4.0 | – | – | – |
| $HangZhou_1$ | Gaussian | 3.5 | 442.82±11.29 | 377.84±4.36 | **365.87±4.86** |
| | U-rand | 3.5 | 495.52±3.41 | 501.25±32.91 | **408.16±11.38** |
| | MinQ | 3.5 | – | – | – |
| | MAD | 3.5 | – | – | – |
| | Gaussian | 4.0 | 453.73±15.11 | 383.04±6.20 | **377.94±8.25** |
| | U-rand | 4.0 | 505.68±11.11 | 532.49±30.07 | **422.86±10.07** |
| | MinQ | 4.0 | – | – | – |
| | MAD | 4.0 | – | – | – |

Table 15: **PSNR Performance Comparison with RobustLight (higher is better)**

| Dataset | Type | Scale | Advanced-CoLight | | |
|---------|------|-------|------|-------------|---------------|
| | | | base | RobustLight | RobustLight++ |
| $JiNan_1$ | Gaussian | 3.5 | 14.25 | 17.67 | **20.87** |
| | U-rand | 3.5 | 7.30 | 11.12 | **16.16** |
| | Mask | 25% | 10.76 | 14.02 | **14.74** |
| | Mask | 50% | 5.04 | **6.93** | 5.49 |
| $JiNan_2$ | Gaussian | 3.5 | 14.25 | 20.00 | **22.58** |
| | U-rand | 3.5 | 7.30 | 11.75 | **17.57** |
| | Mask | 25% | 22.08 | **28.65** | 28.07 |
| | Mask | 50% | 5.80 | **21.53** | 17.67 |
| $HangZhou_2$ | Gaussian | 3.5 | 14.32 | 15.88 | **16.55** |
| | U-rand | 3.5 | 7.26 | **12.16** | 11.98 |
| | Mask | 25% | 6.30 | 6.32 | **6.71** |
| | Mask | 50% | 3.01 | **4.25** | 4.17 |

and single epoch settings of 25 and 90. The TSC RL agent is configured with a discount factor of 0.8, a buffer capacity of 12,000, a batch size of 20, and an Adam optimizer with a learning rate of 0.001, incorporating an epsilon greedy strategy with initial, minimum, and decay values of 0.8, 0.2, and 0.95, respectively.

Table 16: **MAE Performance Comparison with RobustLight (lower is better)**

| Dataset | Type | Scale | Advanced-CoLight | | |
|---|---|---|---|---|---|
| | | | base | RobustLight | RobustLight++ |
| $JiNan_1$ | Gaussian | 3.5 | 4.44 | 2.52 | **2.35** |
| | U-rand | 3.5 | 5.31 | 5.17 | **2.92** |
| | Mask | 25% | 1.50 | **1.00** | 1.11 |
| | Mask | 50% | 1.22 | 1.63 | 1.45 |
| $JiNan_2$ | Gaussian | 3.5 | 8.38 | 4.55 | **4.11** |
| | U-rand | 3.5 | 5.28 | 4.90 | **4.75** |
| | Mask | 25% | 1.00 | 0.76 | **0.74** |
| | Mask | 50% | 1.77 | 1.52 | **1.19** |
| $HangZhou_2$ | Gaussian | 3.5 | 1.43 | **1.15** | 1.16 |
| | U-rand | 3.5 | 2.92 | 1.86 | **1.64** |
| | Mask | 25% | 0.80 | 0.76 | **0.72** |
| | Mask | 50% | 1.37 | 1.34 | **0.99** |

Table 17: **Inference time comparison (in milliseconds) based on Advanced-Colight.**

| $JiNan_1$ | | | $HangZhou_1$ | | |
|---|---|---|---|---|---|
| Type | RobustLight | **Our** | Type | RobustLight | **Our** |
| Gaussian | 131.52 | **33.40** | Gaussian | 139.60 | **31.89** |
| U-rand | 173.76 | **38.23** | U-rand | 179.18 | **36.24** |
| MAD | 1049.95 | **119.19** | MAD | 1505.61 | **162.36** |
| MinQ | 1081.03 | **118.96** | MinQ | 1524.30 | **159.83** |
| Mask 25% | 612.33 | **72.57** | Mask 25% | 1095.14 | **75.18** |
| Mask 50% | 997.53 | **86.39** | Mask 50% | 1095.11 | **74.52** |

Table 18: **Hyperparameters**

| Hyperparameter Type | Hyperparameter | Setting |
|---|---|---|
| UNet/Model Hyperparameter | embed_dim | 64 |
| | state_dim | 20/32 |
| | action_dim | 4 |
| | hidden_dim | 256 |
| Diffusion Training Hyperparameter | non_markovian_step | 6 |
| | condition_length | 4 |
| | beta schedule | 3.0651, 24.552, -3.1702 |
| | discount($\gamma$) | 0.99 |
| | target critic($\tau$) | 0.005 |
| | diffusion timestep | 100 |
| | batch size | 16 |
| | buffer capacity | 240 |
| | optimizer | Adam |
| | learning rate | 0.0003 |
| | meta learning rate $\mu$ | 0.1 |
| | epochs(meta/single) | 25 / 90 |
| | hidden size | 256 |
| | attention embed_dim | 64 |
| TSC RL Agent Training Hyperparameter | discount($\gamma$) | 0.8 |
| | buffer capacity | 12000 |
| | epochs | 80 |
| | batch_size | 20 |
| | learning_rate | 0.001 |
| | target update time | 5 |
| | normal factor | 20 |
| | loss function | mean_squared_error |
| | optimizer | Adam |
| | learning rate | 0.001 |
| | patience | 10 |
| | epsilon (init/min/decay) | 0.8 / 0.2 / 0.95 |
| | D_DENSE | 20 |

