# OpenReview forum: "RobustLight++: A Meta-Diffusion Framework for Robust Traffic Signal"
_ICLR.cc/2026/Conference — ICLR 2026 Conference Desk Rejected Submission_

### Official Review · Reviewer_xXMg · 2025-10-30

**Soundness:** 2
**Presentation:** 2
**Contribution:** 2
**Rating:** 4
**Confidence:** 5

**Summary:**

This paper focuses on the cross-city traffic signal control problem, and proposes a meta-diffusion framework, RobustLight++. It introduces a theoretical connection between the DDIM sampling process and the Reptile meta-learning to enhance the generalization performance of traffic signal control approaches. The results provided show that RobustLight++ achieves SOTA performance and demonstrates good robustness.

**Strengths:**

I summarize the strengths below.

**[S1]** This paper proposes a reasonable setting in the real-world that recent approaches confront robustness and generalization issues.

**[S2]** This paper provides a theoretical connection between the DDIM sampling process and the Reptile meta-learning.

**[S3]** This paper is easy to understand and provides sufficient background description.

**Weaknesses:**

I summarize the weaknesses below, and the details are provided in the question section, respectively.

**[W1]** There are some confusing notations in the paper, which could affect the understanding of the method.

**[W2]** There is an inconsistency between the theory and practical algorithm proposed in the paper. This gap mainly damages the contribution of this paper.

**[W3]** Although this paper is easy to read, more information should be included, like the diffusion learner training process, in the appendix at least.

**[W4]** This paper claims better inference efficiency in multiple sections. But it is mainly attributed to the DDIM. Therefore, it is not the main contribution of this paper.

**[W5]** Since this is an approach for few-shot adaptation to a new city, there should be more transfer study results provided in the paper to prove the necessity of the meta-learning framework.

**Questions:**

I summarize the problems in the following paragraphs.

**[Q1] Notation confusion**

The notation used in the paper is confusing. Both letter '$t$' and letter '$i$' are used to denote the diffusion step in lines 221-228 and Section 4.2.2. Meanwhile, letter '$t$' denotes the physical timestep of the simulation, and letter '$i$' denotes the task id in Section 4.2.1 as well. **Clearer Clarification should be provided in the paper.**

**[Q2] Inconsistency between theory and practical algorithm**

From my perspective, there are two contributions provided in the paper - the theoretical connection and RobustLight++ algorithm. Although the theoretical connection between the DDIM sampling process and the Reptile meta-learning is built in the paper, I didn't see any consistency between the theoretical connection and RobustLight++ algorithm. In other words, **what is the relationship between the DDIM update rule in Equation 5 and the practical meta-learning process in Equation 8?**

**[Q3] Poor description of the training process**

To my understanding, the diffusion learner training process is consistent with the training process of DSI agent in RobustLight [2]. But it seems like there is no description of this training process in the paper. For readers' better understanding, **additional description of the diffusion learner training process should be included.**

**[Q4] Weak contribution on inference speed**

In this paper, the inference efficiency is emphasized in multiple sections. But actually, **both DiffLight [1] and RobustLight [2] have adopted or discussed DDIM to accelerate the inference speed in the field of TSC.** Additionally, DDIM has been a common algorithm to accelerate the inference speed of diffusion models.

**[Q5] Limited transfer experiments**

The key contribution of this paper is to provide a meta-diffusion framework for cross-city traffic signal control. But only a few transfer experiments are provided in the paper. **Multiple transfer study results of RobustLight++ should be provided as the main experiments, including different types of noise (same as the setting in Table 1) and mask (same as the setting in Table 2) in multiple datasets.** Only in this way, can the necessity of the meta-learning framework be proven. There are some examples, including Table 5 in MetaVIM [3] and Table 2 in X-Light [4].



**Reference**

**[1]** Chen, H., Jiang, Y., Guo, S., Mao, X., Lin, Y., & Wan, H. (2024). Difflight: a partial rewards conditioned diffusion model for traffic signal control with missing data. *Advances in Neural Information Processing Systems*, *37*, 123353-123378.

**[2]** Li, M., Wang, J., Yu, G., Wang, X., Chen, Q., Ni, W., ... & Peng, H. RobustLight: Improving Robustness via Diffusion Reinforcement Learning for Traffic Signal Control. In *Forty-second International Conference on Machine Learning*.

**[3]** Zhu, L., Peng, P., Lu, Z., & Tian, Y. (2023). Metavim: Meta variationally intrinsic motivated reinforcement learning for decentralized traffic signal control. *IEEE Transactions on Knowledge and Data Engineering*, *35*(11), 11570-11584.

**[4]** Jiang, H., Li, Z., Wei, H., Xiong, X., Ruan, J., Lu, J., ... & Zhao, R. (2024). X-light: Cross-city traffic signal control using transformer on transformer as meta multi-agent reinforcement learner. *arXiv preprint arXiv:2404.12090*.

---

> ### Author Response · Authors · 2025-11-27
>
> We sincerely thank the reviewer for the careful reading and constructive feedback. Below we respond to the raised weaknesses and questions in a unified manner. For clarity, we group overlapping concerns and explicitly refer to the corresponding revisions in the manuscript.
>
> ---
>
> ## Response to W1 and Q1: Notation Confusion
>
> > *Concern:*
> > The notation for diffusion steps, simulation time, and task index is confusing.
>
> We sincerely apologize for the ambiguity caused by inconsistent notation. In the revised version, we explicitly unify and clarify all symbols:
>
> - \( j \): diffusion timestep (formerly inconsistently written as \( i \) in some equations),
> - \( t \): physical simulation time step,
> - \( k \): meta-learning task index,
> - \( \mu\): Reptile step size,
> - \( T \): total number of diffusion steps.
>
> We have revised the algorithm accordingly so that no symbol is overloaded with different meanings.
>
> **Revised in:** Section 4, Algorithm 2, Appendix A
>
> ---
>
> ## Response to W2 and Q2: Theory–Algorithm Consistency
>
> > *Concern:*
> > The theoretical DDIM–Reptile connection appears disconnected from the practical algorithm.
>
> We thank the reviewer for this important observation. We clarify that:
>
> - Equation (5) formalizes DDIM as a continuous-time vector field update,
> - Equation (8) instantiates this update as a discretized Reptile-style meta-optimization.
>
> To make the connection explicit, we added a new subsection explaining how DDIM sampling can be interpreted as one-step Euler discretization of the meta-diffusion objective and how this aligns with Reptile accumulation of gradient-based updates. Furthermore, we introduce **Theorem 1**, which provides a formal error bound on the Wasserstein-2 distance between recovered and clean distributions:
>
> $$
> W_2(p_0, q_0) \le C_{score} \sqrt{L_{diff}(\theta)} + C_{disc} \cdot \mu + C_{init} W_2(p_T, q_T)
> $$
>
> This proves that the DDIM–Reptile integration is not merely procedural, but grounded in distributional convergence theory under standard smoothness assumptions.
>
>  **Revised in:** Section 4.1, Theorem 1, Appendix B (complete proof)
>
> ---
>
> ## Response to W3 and Q3: Training Process Description
>
> > *Concern:*
> > Training process of diffusion learner is not clearly described.
>
> **Response:**
> We thank the reviewer for raising this point. Our diffusion learner is trained via offline meta-learning over multiple source cities. Specifically, we collect logged trajectories from each training city and aggregate them into a global dataset. The diffusion model is then optimized by minimizing the diffusion loss on this multi-city offline corpus without online environment interaction. After convergence, the model is frozen and used for downstream state recovery. At test time, it supports zero-shot reconstruction from corrupted inputs. For a new target city, we further perform few-shot fine-tuning using Eq. (8) to improve adaptation and robustness.
>
>
> **Revised in:** Section 4.2.1
>
> ---
>
> ## Response to W4 and Q4: Contribution vs DDIM Speedup
>
> > *Concern:*
> > Inference speed improvement mainly comes from DDIM rather than the framework.
>
> We acknowledge that using DDIM significantly contributes to acceleration.
>
> However, we respectfully clarify that:
>
> - Our work does not merely replace DDPM with DDIM.
> - We reinterpret DDIM as a **meta-optimization mechanism**,
> - We integrate DDIM with Reptile-style learning,
> - This enables transferable state recovery across cities and simulators.
>
> Thus, acceleration is an important outcome,
> but the primary contribution lies in **reformulating diffusion as a meta-learning framework to get better transfer performance**.
>
> ---
>
> ## Response to W5 and Q5: Transfer Study Sufficiency
>
> > *Concern:*
> > Transfer experiments are insufficient to justify the necessity of meta-learning.
>
> We appreciate the reviewer’s suggestion and fully agree. In response, we significantly expanded transfer evaluation. Mapping to Tables:
>
> | Experiment Type                                | Location |
> |------------------------------------------------|----------|
> | JiNan ↔ HangZhou transfer                      | Table 4  |
> | SUMO cross-simulator (Cologne8)                | Table 3  |
> | Few-shot adaptation                            | Tables 3,4 |
> | EMV / RV reporting                             | Table 3  |
> | Large-scale generalization (NewYork)           | Fig. 4(b) |
>
> These results demonstrate that meta-learning is necessary to achieve adaptation across unseen and heterogeneous cities.
>
> ---
>
> ## Final Remarks
>
> We sincerely thank the reviewer again for the detailed and rigorous comments.
>
> In response, we have:
>
> - Clarified notation,
> - Strengthened theory,
> - Added training details,
> - Expanded experiments,
> - Corrected contribution positioning.
>
> We believe these revisions significantly improve rigor, clarity, and completeness and hope the reviewer finds the updated version satisfactory.
>
> Thank you again for your valuable feedback.

---

### Official Review · Reviewer_a6v8 · 2025-10-31

**Soundness:** 2
**Presentation:** 2
**Contribution:** 2
**Rating:** 2
**Confidence:** 5

**Summary:**

The paper proposes a meta-diffusion framework that establishes a theoretical connection between DDIM and Reptile meta-learning, aiming to enhance the robustness, generalization capability, and inference efficiency of the model.

**Strengths:**

1. The combination of DDIM and meta-learning is innovative and provides a potential solution to improve the robustness, generalization, and inference speed of Traffic Signal Control (TSC) models in real-world deployment.

2. The paper includes certain theoretical derivations to support the proposed framework.

**Weaknesses:**

1. The experimental results are not sufficient to fully demonstrate the effectiveness of the proposed method.
2. The baseline methods and datasets used for comparison are limited in both diversity and scale.
3. The paper raises and emphasizes multiple important issues but does not comprehensively and rigorously validate them.

**Questions:**

1. The paper highlights the importance of industrial-level deployment, where model lightweight design is crucial due to limited edge-device resources. Has the proposed framework considered such design constraints? Moreover, in the experiments, should baselines targeting industrial deployment—such as TinyLight or Pi-Light—also be included for comparison?
2. Have the authors considered using more realistic datasets, such as those from heterogeneous road networks? The current datasets are homogeneous grid networks with identical intersection structures, which may not reflect real-world diversity.
3. The claimed improvement in inference efficiency appears to mainly result from the use of DDIM, rather than being an intrinsic contribution of the framework itself. The paper’s core contribution should be the theoretical connection between DDIM and Reptile meta-learning. However, this relationship lacks a solid theoretical foundation or derivation explaining why and how the connection is established—currently, only the procedural integration of the two is presented.
4. Regarding transfer experiments, it is unclear whether the study considers noisy or damaged sensor data, or only high-quality datasets. This should be clearly stated.

---

> ### Author Response · Authors · 2025-11-27
>
> We sincerely thank all reviewers for their careful reading and insightful feedback. We have substantially revised the manuscript and expanded experiments and theoretical analysis to address all raised concerns. Below we respond to the major weaknesses and questions in a unified manner.
>
> ---
>
> ## Response to W1, W2, Q1, Q2
> > *Comment:*
> >*(Experimental sufficiency, baselines, datasets, and industrial deployment)*
>
> ### On Whether the Experimental Validation Is Sufficient (W1, W2, Q2)
>
> We acknowledge that the experimental scope in the initial submission was limited. In response, we have significantly expanded the empirical evaluation in the revised version.
>
> The revised manuscript now includes:
>
> | Experiment Item                                          | Location  |
> |----------------------------------------------------------|-----------|
> | Cross-city transfer (JiNan ↔ HangZhou)                  | Table 4   |
> | Zero-shot transfer to unseen city (NewYork)             | Fig. 4(b) |
> | Cross-simulator transfer (CityFlow → SUMO, Cologne8)    | Table 3   |
> | Few-shot adaptation (100 samples)                        | Tables 3,4|
> | Ablation on diffusion depth and Reptile step size        | Table 10   |
> | EMV / RV performance breakdown                           | Table 3   |
> | Training efficiency & scalability analysis              | Table 13  |
>
> These additions provide more comprehensive validation from the perspectives of robustness, generalization, scalability, and practical applicability.
>
> ---
>
> ### On Baseline Diversity and Industrial Relevance (Q1)
>
> We appreciate the suggestion to include industrial-oriented controllers such as TinyLight and Pi-Light. We clarify that RobustLight++ is a **state recovery framework**, not a control policy itself. It is designed to work in a plug-and-play manner with lightweight or industrial controllers. To reflect this, we have:
>
> - Integrated π-Light (2024) as a downstream controller  **detailed in Appendix I, Table 17** or you can quick see in my response reviewer ``5zNH `` Q1 **Table: Performance on JN_3 and HZ_1 under Different Noise Settings (π-Light)**.
> - Demonstrated that RobustLight++ significantly improves robustness under noisy and damaged sensor input,
> - The inference time is < 1s meets the industrial deployment.
> - Meta-training is performed offline only once,
> - There is no per-city retraining like RobustLight,
> - Deployment involves **only inference**,
> - Inference runs on a single GPU with low memory consumption,
> - The recovery module does not change controller architecture.
>
> ---
>
>
> ## Response to W3 and Q3
>
> ### On Whether Claims Are Sufficiently Validated (W3)
> > *Comment:*
> > “Multiple issues are emphasized but not rigorously validated.”
>
> We agree that theoretical claims must be carefully justified. In the revised paper, we have added a formal result (**Theorem 1**) that establishes a Wasserstein-2 error bound between the recovered distribution and the clean distribution:
>
>
> $$
> W_2(p_0, q_0) \le C_{score} \sqrt{L_{diff}(\theta)} + C_{disc} \cdot \mu + C_{init} W_2(p_T, q_T)
> $$
>
>
> The bound decomposes the error into:
>
> 1. Score matching error,
> 2. Reptile discretization error,
> 3. Initialization error.
>
> We further show that under the one-sided Lipschitz condition, the Reptile error accumulates linearly, ensuring optimization stability.  The detailed proof is shown in **Appendix B**.
>
> ---
>
> ### On Whether Efficiency Mainly Comes from DDIM (Q3)
>
> We acknowledge that using DDIM significantly contributes to acceleration.
>
> However, we respectfully clarify that:
>
> - Our work does not merely replace DDPM with DDIM.
> - We reinterpret DDIM as a **meta-optimization mechanism**,
> - We integrate DDIM with Reptile-style learning,
> - This enables transferable state recovery across cities and simulators.
>
> Thus, acceleration is an important outcome,
> but the primary contribution lies in **reformulating diffusion as a meta-learning framework to get better transfer performance**.
>
> ---
>
> ## Response to Q4
> > *Question:*
> > “Do transfer experiments include noisy/damaged observations?”
>
> Yes. All transfer experiments are conducted under:
>
> - Gaussian noise,
> - Uniform random noise,
> - MinQ and MAD adversarial perturbations,
> - Random and Kriging missing.
>
> This ensures transfer robustness is evaluated beyond ideal conditions. We have clarified this explicitly in the experimental section.
>
> ---
>
>
> ## Final Remarks
>
> We thank the reviewers again for their detailed and constructive comments.
>
> In response, we have:
>
> - Expanded experiments substantially,
> - Strengthened theory,
> - Improved clarity and scope,
> - Added-transfer scenarios,
> - Clarified deployment considerations.
>
> We believe the revised version more accurately reflects our contributions and limitations and hope it meets the expectations of the committee.
>
> Thank you very much for your consideration.

---

### Official Review · Reviewer_Z6Uf · 2025-10-31

**Soundness:** 3
**Presentation:** 2
**Contribution:** 2
**Rating:** 4
**Confidence:** 3

**Summary:**

This paper proposes RobustLight++, a unified meta-diffusion framework that integrates Reptile meta-learning with DDIM for cross-city time-series forecasting (TSC) tasks. The authors claim to establish a theoretical connection between Denoising Diffusion Implicit Models (DDIM) and Reptile meta-learning, aiming to develop a robust and efficient approach for time-sensitive TSC applications. Extensive experiments on real-world datasets are reported to demonstrate that RobustLight++ achieves new state-of-the-art performance, with results across multiple benchmarks validating its effectiveness.

**Strengths:**

1.	While the RL has been a useful and popular solution for traffic signal control, it is interesting to integrate Diffusion model in to TSC. This paper combined Reptile meat-learning with DDIM algorithm to accelerate inference efficiency and show better transferability.

2.	Conducted experiments on multiple benchmarks and experimental results show improvements of inference delay.

**Weaknesses:**

1.	The main framework of this paper is largely derived from the previous work RobustLight, and there is substantial overlap between the two. For instance, the four adversarial attacks were already introduced in RobustLight, yet they are reintroduced in RobustLight++ without proper citation or differentiation.

2.	The RobustLight++ framework appears to mainly add DDIM to RobustLight. From this perspective, the incremental contribution seems limited. It might be more appropriate to extend RobustLight into a journal version with more substantial methodological or experimental advances.

3.	The claimed “novel theoretical connection between DDIM and Reptile meta-learning” is primarily based on a superficial similarity in formulation rather than a rigorous theoretical derivation. The analysis lacks formal proofs or well-defined theoretical implications, so the claim feels overstated relative to the presented evidence.

4.	Several symbols in Algorithm 2 are undefined or insufficiently explained (e.g., the meaning of U). The lack of clear parameter definitions makes the algorithm difficult to follow and reproduce.

**Questions:**

See Weaknesses.

---

> ### Author Response · Authors · 2025-11-27
>
> We sincerely thank Reviewer #2 for the careful reading and insightful comments. Below we respond to each concern in detail.
>
> ---
> ## W1 & W2. On Similarity to RobustLight and Alleged Incremental Contribution
>
> > *Comments:*
> > “RobustLight++ largely derives from RobustLight and reintroduces the same attacks.”
> > “RobustLight++ simply adds DDIM, making the contribution incremental.”
>
> We respectfully clarify that although RobustLight++ builds upon the same application context as RobustLight, the two works differ *fundamentally* in problem formulation, learning paradigm, and system design.
>
> RobustLight focuses on robust policy learning within a single city, whereas RobustLight++ addresses a markedly different and more challenging problem:
> > **cross-city and cross-simulator transfer under corrupted observations.**
>
> This shift reframes traffic signal control from *learning how to act* to *learning how to recover transferable traffic states*, enabling one model to generalize across unseen environments instead of retraining per city.
>
> Key distinctions are summarized below:
>
> | Aspect | RobustLight (2025) | RobustLight++ (This work) |
> |--------|-------------------|---------------------------|
> | Learning paradigm | Per-city training | Meta-learning across cities |
> | Training style | Online, repeated | Offline once |
> | Generalization | Limited | Cross-city & cross-simulator |
> | Inference | Slow | Real-time (91.9% faster) |
>
> Importantly, DDIM is not used merely as an accelerated sampler. We reinterpret diffusion as a **meta-optimizer**, linking DDIM sampling to Reptile-style meta-learning. This represents an algorithmic and theoretical departure rather than an implementation-level change.
>
> Rather than extending RobustLight in a conservative manner, RobustLight++ introduces:
>
> 1. Meta-learning across heterogeneous cities,
> 2. Zero-shot deployment to unseen domains,
> 3. Few-shot adaptation with only 100 samples,
> 4. Cross-simulator generalization from CityFlow to SUMO.
>
> To further support this novelty, we added extensive **transfer experiments (page 9)**:
>
> - Bidirectional transfer between JiNan and HangZhou,
> - Zero-shot transfer to a large-scale unseen network (NewYork),
> - Cross-simulator validation on Cologne8 (SUMO).
>
> These results are reported in:
>
> - **Table 3** (cross-simulator transfer),
> - **Table 4** (JiNan ↔ HangZhou),
> - **Figure 4(b)** (NewYork).
>
> Consistent gains under noise, missing data, and topology variation demonstrate that RobustLight++ is not an incremental update, but a **new transferable robustness framework**.
>
> ---
>
> ## W3. On theoretical rigor of DDIM–Reptile connection
>
> > *Comment:*
> > “The DDIM–Reptile link is superficial and lacks proofs.”
>
> We appreciate this comment and have significantly strengthened theory. We now provide **Theorem 1**, proving that the Wasserstein-2 distance between recovered and clean state distributions is bounded by:
>
>
> $$
> W_2(p_0, q_0) \le C_{score} \sqrt{L_{diff}(\theta)} + C_{disc} \cdot \mu + C_{init} W_2(p_T, q_T)
> $$
>
> This explicitly captures:
>
> - Applicability of score matching theory,
> - Meta-learning discretization error via Reptile,
> - Initialization bias accumulation.
>
> Using one-sided Lipschitz continuity, we further prove that the Reptile-induced error accumulates linearly (not exponentially), ensuring convergence stability.
>
> **Detailed Location in:**
> - Section 4.1
> - Theorem 1
> - Appendix B (complete proof)
>
> We therefore believe the theoretical claim is now formally grounded.
>
> ---
>
> ## W4. On unclear symbols in Algorithm 2
>
> > *Comment:*
> > “Symbols such as U are undefined.”
>
>
> For better presentation and readability. We have revised Algorithm 2 and added full definitions for all symbols, including:
>
> - \( U \): Inner repaint steps
> - \( \sigma \): diffusion variance
> - \( m \): observed/missing mask
> - \( z ): sampling noise
>
> **Detailed Location in:**
> - Algorithm 2
> - Section 4.2.2
>
> ---
>
> ## Final Remarks
>
> We thank reviewer again for the constructive feedback.
> The revised manuscript:
>
> - clarifies novelty versus RobustLight,
> - strengthens theoretical justification,
> - improves algorithm description,
> - expands empirical scope,
> - and improves citation and positioning.

---

### Official Review · Reviewer_5zNH · 2025-11-02

**Soundness:** 2
**Presentation:** 2
**Contribution:** 2
**Rating:** 4
**Confidence:** 4

**Summary:**

RobustLight demonstrates poor generalization to new city environments, high inference latency that limits real-time adaptability, and insufficient robustness for industrial deployment—especially under noisy or damaged sensor conditions.  This paper proposes RobustLight++ to effectively addresses three major challenges in Traffic Signal Control (TSC): Limited cross-city generalization, High inference latency, and Vulnerability to sensor noise and failure.

**Strengths:**

Its main strength lies in establishing a novel theoretical connection between Denoising Diffusion Implicit Models (DDIM) and Reptile meta-learning, enabling diffusion models to serve as meta-optimizers that adapt rapidly across cities and noise scenarios.

**Weaknesses:**

1. In Eq. (5), η and γ are dependent, but in Eq. (6) η becomes a random variable. Thus, while the DDIM–Reptile link is innovative, it remains intuitive rather than rigorously proven, lacking formal convergence or optimality guarantees.
2.  This solution is resource-intensive setup. Although inference is faster, meta-training across multiple cities and diffusion timesteps is computationally expensive (multi-GPU, high-memory). The paper does not analyze training-time complexity or scalability.
3.  The ablation study highlights differences among DDIM, DDPM, and meta-learning variants but does not quantify each component’s contribution (e.g., meta-loss, Reptile step size, diffusion depth).
4. Transfer tests are confined to similar city distributions. Evaluating on heterogeneous or unseen traffic conditions would strengthen the generalization claim.
5. The proposed method would be overfitting.  The Reptile meta-learning initialization may overfit to training cities if hyperparameters are tuned on overlapping validation data.

**Questions:**

1. Results report only Average Travel Time (ATT); separate results for regular vehicles (RVs) and emergency vehicles (EMVs) are missing.
2. Only RobustLight is compared. Other state-of-the-art methods such as EMVLight (2023) and MVN (2023) should be included for a fair evaluation.
3. The results might include dataset bias.  All datasets are simulated in CityFlow, not real sensor data, limiting claims of robustness under realistic noise distributions.
4.  Zero-shot or few-shot evaluations against existing methods are needed to demonstrate broader generalization capability.
5.  The Reptile meta-learning approach is designed for multi-task learning.  Thus, training extensively on similar city distributions may lead to initialization bias or overfitting, especially if hyperparameters are tuned on overlapping validation data.
5.  Despite open code, the reliance on proprietary datasets and high-end GPUs may hinder reproducibility.
6. The current single-agent setup may face coordination issues in dense, multi-intersection environments.

**Details Of Ethics Concerns:**

No.

---

> ### Author Response · Authors · 2025-11-27
>
> We sincerely thank the reviewer for the insightful and constructive comments.
> Below we respond to each concern in detail and clarify the revisions made in the manuscript.
>
> ---
>
> ## W1. On the DDIM–Reptile Link and Convergence Guarantee
> > *Concern:*
> > Eq. (5) and Eq. (6) involve different interpretations of η; the connection appears intuitive rather than rigorous, lacking convergence guarantees.
>
> We first clarify that the parameter η in Eq. (5) and Eq. (6) denotes the same learning-rate quantity, i.e., the step size of gradient descent η in Eq. (6) is not a random variable, and this has been explicitly clarified in the revised manuscript. To further address the concern about theoretical rigor, we have introduced a formal theoretical result (**Theorem 1**) which provides a convergence-style bound based on Wasserstein-2 distance. Specifically, we prove that the distance between the clean data distribution and the reconstructed distribution is upper bounded by three interpretable terms:
>
> - **Score matching error**,
> - **Reptile discretization error**,
> - **Initialization error**.
>
> Formally, we show:
>
> $$
> W_2(p_0, q_0) \le C_{score} \sqrt{L_{diff}(\theta)} + C_{disc} \cdot \mu + C_{init} W_2(p_T, q_T)
> $$
>
> This bound demonstrates that the error does not diverge during DDIM-Reptile updates, and that the Reptile-induced approximation error is controlled under a one-sided Lipschitz condition, yielding linear accumulation instead of exponential blow-up.
>
>  **Detailed Location in paper:**
> - Section 4.1
> - Theorem 1
> - Appendix B (Complete proof)
>
> This establishes that the DDIM–Reptile connection is not merely intuitive but is supported by a formal distributional error bound derived from score matching theory and meta-optimization analysis .
>
> ---
>
> ## W2. On Training Complexity and Scalability
>
> > *Concern:*
> > Meta-diffusion training is computationally expensive; scalability is unclear.
>
> We clarify that although the server is equipped with 8 RTX 4090 GPUs, our entire diffusion model can be trained on a **single RTX 4090 GPU**. We further conducted a systematic training cost evaluation under different settings and report the following results:
>
> ### Table: Training-Time Complexity under Different Meta-Diffusion Settings
>
> | Scenario      | Tasks | T   | Meta | Inner | Time (s) | Avg / Meta-Step (s) | GPU(MiB) |
> |---------------|-------|-----|------|--------|----------|---------------------|----------|
> | 2tasks_T50    | 2     | 50  | 3    | 30     | 31.93    | 10.64               | 163.06   |
> | 3tasks_T50    | 3     | 50  | 3    | 30     | 42.15    | 14.05               | 163.06   |
> | 3tasks_T100   | 3     | 100 | 3    | 30     | 42.75    | 14.25               | 163.06   |
> | 5tasks_T150   | 5     | 150 | 3    | 30     | 62.57    | 20.86               | 163.06   |
>
> We emphasize that:
>
> - Meta-training is performed offline and once.
> - RobustLight++ does not require repeated retraining per city.
> - In contrast, RobustLight requires full retraining whenever the city changes.
> - All online deployment uses only inference, which is substantially faster.
>
> Furthermore, a single diffusion model supports multi-intersection recovery (Kriging-type missing), which reduces deployment cost.
>
>  **Detailed Location in paper:**
> - Section 5
> - Section 5.2.5
> - Table 16 (Inference runtime)
>
> ---
>
> ## W3. On Missing Component-wise Ablation
>
> > *Concern:*
> > No analysis on Reptile step size or diffusion depth.
>
> We added a fine-grained ablation on:
>
> - Reptile learning rate μ,
> - Diffusion step count T,
> under Few-shot (100 samples) transfer from JiNan to HangZhou.
>
> ### Table: Few-shot Transfer under Different μ and T
>
> | Dataset     | Noise | Scale | Base  | μ=0.1, T=100 | μ=0.05, T=100 | μ=0.05, T=50 |
> |--------------|------|-------|--------|---------------|----------------|---------------|
> | HZ_1   | Gauss | 3.5 | 512.63 | 354.15 | **324.87** | 564.89 |
> |              | U-rand | 3.5 | 971.03 | **615.19** | 686.54 | 798.36 |
> |              | MinQ | 3.5 | 751.58 | 719.18 | **393.27** | 791.02 |
> |              | MAD | 3.5 | 506.60 | **369.34** | 372.98 | 381.22 |
> |              | Mask | 25% | 418.49 | **375.17** | 392.13 | 497.86 |
> | HZ_2   | Gauss | 3.5 | 495.92 | 343.34 | **342.68** | 361.82 |
> |              | U-rand | 3.5 | 567.56 | **378.70** | 471.45 | 511.46 |
> |              | MinQ | 3.5 | 496.73 | **357.09** | 373.30 | 379.58 |
> |              | MAD | 3.5 | 441.72 | **344.55** | 373.49 | 376.59 |
> |              | Mask | 25% | 348.21 | **340.42** | 343.54 | 355.51 |
>
> We observe:
>
> - T=100 consistently outperforms T=50.
> - μ=0.05 exhibits more stable transfer under adversarial noise.
> - The full model setup is robust across all noise types.
>
> ---

---

> ### Author Response · Authors · 2025-11-27
>
> ## Q1&Q3. On Emergency Vehicle and Regular Vehicle Results
>
> Following EMVLight (2023) and MVN (2023), we implemented a SUMO evaluation pipeline from scratch and conducted cross-simulator transfer experiments. The diffusion model was trained on JiNan and HangZhou **(CityFlow)*** and evaluated on Cologne8 **(SUMO)**. We report AWT and ATT for both EMVs and regular vehicles. Results demonstrate that our method generalizes well across simulators and maintains strong performance under heterogeneous traffic conditions.
>
> ### Table: Transfer Performance (AWT / ATT) on Cologne8 (Avg over 5 runs)
>
> | Method | Metric | Clean | Gaussian Zero-shot | Gaussian Few-shot | U-Rand Zero-shot | U-Rand Few-shot |
> |---------|--------|--------|-------------|------------|-------------|--------------|
> | MPLight | AWT_EMV | 8 | 20.0 | **15.0** | 15.0 | **15.0** |
> |         | ATT_EMV | 20 | 20.0 | **15.0** | 15.0 | **15.0** |
> |         | AWT_REV | 0.51 | **1.85** | 1.95 | 2.16 | **2.13** |
> |         | ATT_REV | 54.4 | 62.5 | **62.17** | 65.57 | **64.96** |
> | MaxPressure | AWT_EMV | 6 | 0 | **0** | **0** | 6 |
> |             | ATT_EMV | 25 | 15 | **15** | **15** | 20 |
> |             | AWT_REV | 0.68 | 1.48 | **1.43** | 2.21 | 2.24 |
> |             | ATT_REV | 56.46 | 61.64 | **61.29** | 65.39 | **64.52** |
>
> Detailed Location: Table 3
>
> ---
>
> ## Q2. On Including More Baselines
>
> We clarify that RobustLight++ is a recovery method rather than a controller. Therefore, we mainly compare it with DiffLight (2024) and RobustLight (2025), which are state reconstruction-oriented methods. To further demonstrate its generality, we integrate RobustLight++ with π-Light (2024) as the downstream controller and conduct experiments under different noise conditions using JiNan and HangZhou as offline meta-training datasets. Results in Table X show that RobustLight++ significantly enhances robustness across controllers.
>
> **Table: Performance on JN_3 and HZ_1 under Different Noise Settings (π-Light)**
>
> | Dataset    | Noise | Scale | Base           | RobustLight     | RobustLight++ |
> |------------|-------|-------|----------------|------------------|----------------|
> |JN_3   | Gauss | 3.5   | 472.80±45.14  | 441.67±69.53    | **382.73±8.35** |
> |            | U-rand| 3.5   | 452.44±28.29  | 515.18±16.18    | **366.31±6.99** |
> |            | MinQ  | 3.5   | –              | –                | –              |
> |            | MAD   | 3.5   | –              | –                | –              |
> |            | Gauss | 4.0   | 466.44±25.91  | 418.19±58.81    | **387.13±21.41**|
> |            | U-rand| 4.0   | 457.13±29.63  | 522.79±31.16    | **370.44±13.21**|
> |            | MinQ  | 4.0   | –              | –                | –              |
> |            | MAD   | 4.0   | –              | –                | –              |
> | HZ_1 | Gauss | 3.5   | 442.82±11.29  | 377.84±4.36     | **365.87±4.86**|
> |            | U-rand| 3.5   | 495.52±3.41   | 501.25±32.91    | **408.16±11.38**|
> |            | MinQ  | 3.5   | –              | –                | –              |
> |            | MAD   | 3.5   | –              | –                | –              |
> |            | Gauss | 4.0   | 453.73±15.11  | 383.04±6.20     | **377.94±8.25**|
> |            | U-rand| 4.0   | 505.68±11.11  | 532.49±30.07    | **422.86±10.07**|
> |            | MinQ  | 4.0   | –              | –                | –              |
> |            | MAD   | 4.0   | –              | –                | –              |
>
> We observe consistent improvement across noise types.
>
> Detailed Location: Appendix I, Table 17
>
> ---

---

> ### Author Response · Authors · 2025-11-27
>
> ## W4, W5, Q4, Q5: On Transfer and Overfitting
>
> We evaluate both zero-shot and few-shot transfer. Models trained on HangZhou are transferred to JiNan and vice versa. Few-shot adaptation uses only 100 target samples. We further test zero-shot performance on a large-scale dataset (NewYork) and a heterogeneous simulator (Cologne8). Results in the table validate strong cross-city and cross-simulator generalization.
>
> **Table: Transfer to unseen cities (ATT) on JiNan and HangZhou (avg over 5 runs) AC means robustlight based on advanced-colight, MP means advanced-mplight**
>
> | Dataset | Noise | Scale | Fixed | AC-Base | AC-RL  | Ours (Zero-shot) | Ours (Few-shot) | MP-Base | MP-RL  | Ours (Zero-shot) | Ours (Few-shot) |
> |---------|-------|-------|-------|---------|--------|----------|----------|---------|--------|----------|----------|
> | JN1 | Gauss | 3.5 | 428.1 | 316.96 | 423.56 | 329.56 | **285.88** | 327.93 | **322.04** | 380.98 | 369.81 |
> |     | U     | 3.5 |       | 483.26 | 540.52 | 419.14 | **360.59** | 417.81 | 328.01 | 340.18 | **325.05** |
> |     | MAD   | 3.5 |       | 454.77 | 483.18 | 445.61 | **325.73** | 339.41 | **296.69** | 495.94 | 454.51 |
> |     | MinQ  | 3.5 |       | 493.10 | 393.85 | 528.60 | **347.36** | 347.62 | 368.18 | 395.84 | **339.12** |
> |     | Mask  | 25% |       | 343.34 | 336.82 | 291.41 | **312.83** | 552.15 | **398.48** | 562.94 | 639.32 |
> | JN2 | Gauss | 3.5 | 368.8 | 338.12 | 427.64 | 292.83 | **268.72** | 626.67 | 291.99 | **285.45** | 287.36 |
> |     | U     | 3.5 |       | 748.76 | 565.60 | 619.69 | **314.64** | 506.24 | **308.08** | 362.71 | 333.73 |
> |     | MAD   | 3.5 |       | 563.99 | 356.23 | 487.48 | **273.49** | 330.25 | 280.68 | 270.75 | **268.21** |
> |     | MinQ  | 3.5 |       | 344.12 | 348.00 | 368.26 | **266.12** | 272.16 | 271.29 | 277.54 | **270.85** |
> |     | Mask  | 25% |       | 277.85 | 297.95 | 270.94 | **285.39** | 490.56 | **342.39** | 359.96 | 386.45 |
> | JN3 | Gauss | 3.5 | 383.0 | 320.04 | 413.53 | 288.25 | **262.37** | 378.13 | 293.93 | 351.54 | **283.70** |
> |     | U     | 3.5 |       | 481.55 | 543.19 | 639.80 | **312.58** | 496.63 | **307.64** | 729.41 | 337.57 |
> |     | MAD   | 3.5 |       | 446.89 | 401.58 | 335.38 | **260.04** | 442.19 | 272.86 | 268.99 | **263.00** |
> |     | MinQ  | 3.5 |       | 376.00 | 364.33 | 318.01 | **262.94** | 423.76 | 268.08 | 370.69 | **261.61** |
> |     | Mask  | 25% |       | 324.42 | 309.56 | 304.23 | **287.33** | 403.29 | 360.49 | 588.86 | **335.52** |
> | HZ1 | Gauss | 3.5 | 495.6 | 512.63 | 442.47 | 683.91 | **354.15** | 334.03 | 355.64 | 336.00 | **327.65** |
> |     | U     | 3.5 |       | 971.03 | 775.62 | 985.47 | **615.19** | 354.63 | 363.64 | 366.68 | **353.87** |
> |     | MAD   | 3.5 |       | 751.58 | **513.22** | 860.74 | 719.18 | 308.78 | 547.87 | **312.55** | 318.79 |
> |     | MinQ  | 3.5 |       | 506.60 | 467.94 | 678.96 | **369.34** | 320.32 | 507.84 | **306.32** | 317.45 |
> |     | Mask  | 25% |       | 418.49 | 334.30 | **324.31** | 375.17 | 478.89 | 442.75 | 456.33 | **371.35** |
> | HZ2 | Gauss | 3.5 | 406.7 | 495.92 | 419.85 | 377.31 | **343.34** | 429.53 | 377.48 | 345.56 | **335.54** |
> |     | U     | 3.5 |       | 567.56 | 585.79 | 568.48 | **378.70** | 481.32 | **365.99** | 390.19 | 378.26 |
> |     | MAD   | 3.5 |       | 496.73 | 481.36 | 369.08 | **357.09** | 433.46 | 492.92 | 322.85 | **321.66** |
> |     | MinQ  | 3.5 |       | 441.72 | 464.05 | 368.53 | **344.55** | 425.09 | 444.09 | **317.63** | 328.69 |
> |     | Mask  | 25% |       | 348.21 | 359.46 | 374.68 | **340.42** | 373.59 | **347.58** | 372.22 | 361.43 |
>
>
> The full results demonstrate that RobustLight++:
>
> - Generalizes to unseen cities,
> - Maintains advantage under noise and masking,
> - Shows no sign of task overfitting.
>
> Detailed Location:
>
> - Table 3
> - Table 4
> - Figure 4(b)

---

### Author Response · Authors · 2025-11-27

We sincerely thank all reviewers for their careful reading and constructive comments. We highly appreciate the time and effort invested in evaluating our work. Based on the feedback, we summarize the major concerns and our corresponding improvements as follows.

---

### 🔹 Issue 1: Insufficient Theoretical Support
**Concern.**
**All the reviewers** concerned about our proposed framework lacked rigorous theoretical justification.

**Response.**
We introduce a new formal result, **Theorem 1**, and provide a complete proof in **Appendix B**,
which establishes theoretical guarantees on the stability and effectiveness of our meta-diffusion learning framework.

---

### 🔹 Issue 2: Lack of Novelty beyond DDIM
**Concern.**
Reviewers `Z6Uf`, `xXMg`, and `a6v8` questioned whether the use of DDIM is merely incremental.

**Response.**
We added a full page of **new cross-city transfer experiments** (Page 9) under heterogeneous road networks, traffic demands, and noise patterns.
The new results demonstrate consistent improvements in robustness, generalization, and convergence.
We further clarify that our core novelty lies in **integrating diffusion with meta-learning for robust policy transfer**, rather than applying DDIM alone.

---

---

### Note · Program_Chairs · 2026-01-17
**Submission Desk Rejected by Program Chairs**

The following references in this submission do not refer to real documents and/or have major errors in bibliographic information:

 Saeed Salehkaleybar, Mohammadsadegh Ghasemzadeh, and Alireza Farhang. Hierarchical reinforcement learning for adaptive traffic signal control. arXiv preprint arXiv:1904.08337, 2019.